# Estimating the total variance explained by whole-brain imaging for zero-inflated outcomes
Junting Ren [1] ✉, Robert Loughnan[2,3], Bohan Xu[3], Wesley K. Thompson[1,3] & Chun Chieh Fan [3,4] ✉

There is a dearth of statistical models that adequately capture the total signal attributed to whole-brain imaging features. The total signal is often widely distributed across the brain, with individual imaging features exhibiting small effect sizes for predicting neurobehavioral phenotypes. The challenge of capturing the total signal is compounded by the distribution of neurobehavioral data, particularly responses to psychological questionnaires, which often feature zero-inflated, highly skewed outcomes. To close this gap, we have developed a novel Variational Bayes algorithm that characterizes the total signal captured by whole-brain imaging features for zero-inflated outcomes. Our *zero-inflated variance* (ZIV) estimator estimates the fraction of variance explained (FVE) and the proportion of non-null effects (PNN) from large-scale imaging data. In simulations, ZIV demonstrates superior performance over other linear models. When applied to data from the Adolescent Brain Cognitive Development[SM] (ABCD) Study, we found that whole-brain imaging features contribute to a larger FVE for externalizing behaviors compared to internalizing behaviors. Moreover, focusing on features contributing to the PNN, ZIV estimator localized key neurocircuitry associated with neurobehavioral traits. To the best of our knowledge, the ZIV estimator is the first specialized method for analyzing zero-inflated neuroimaging data, enhancing future studies on brain-behavior relationships and improving the understanding of neurobehavioral disorders.

Non-invasively quantifying the form and function of the human brain and mapping these properties to neurobehavioral outcomes has been fundamental for understanding complex human behaviors[1]. For example, to understand the neural mechanisms underlying the problematic behaviors among youth, researchers often evaluate psychopathology by administering validated questionnaires and measure brain structure and function via magnetic resonance imaging (MRI)[2,3]. Brain regions relevant for psychopathology are inferred by examining association patterns between questionnaire responses and MRI-derived measures[4,5]. For example, it may be determined that the surface area of the prefrontal cortex is associated with the attention domain from a questionnaire, hence leading to the inference that one of the roles of that brain region is to regulate attention[6].

However, inferences based on finding "significant" brain-behavior relationships from a massive univariate analysis approach has been placed in doubt, especially in large studies where effect sizes of individual features can be quite small and the potential for confounding bias can be quite high[7]. For

example, a recent large study found that the top one percent of associations, after an exhaustive search through all functional measures of brain regions, explained at most 0.36% of the variation in cognition[8]. Even though they were statistically significant, the limited variance explained casts doubt on the validity and reliability of the "one-brain-feature-at-a-time" approach[5,7,9,10].

As an additional complication, many neurobehavioral outcomes are *semi-continuous*, i.e., characterized by a peak of values occurring at a minimum value along with typically right-skewed continuous values[11,12]. (Note, following the literature we refer to these distributions as being "zero-inflated" even though the minimum may differ from zero.) Semi-continuous data arise frequently in biomedical applications, including medical costs[13], microbiome[14], single-cell gene expression[15] and psychological questionnaires[16]. For example, the Child Behavior Checklist (CBCL), a widely used assessment of the mental state of children[17], contains eight syndromal subscales and six DSM-oriented subscales, which typically yield

[1]Division of Biostatistics, Herbert Wertheim School of Public Health and Human Longevity Science, University of California San Diego, 9500 Gilman Street, La Jolla 92093 CA, USA. [2]Center for Human Development, University of California San Diego, 9500 Gilman Drive, La Jolla CA, USA. [3]Center for Population Neuroscience and Genetics, Laureate Institute for Brain Research, 6655 S Yale Ave, Tulsa 74136 OK, USA. [4]Department of Radiology, School of Medicine, University of California San Diego, 9500 Gilman Drive, La Jolla 92093 CA, USA. ✉e-mail: junting.ren.stat@gmail.com; CFan@laureateinstitute.org

right-skewed data with evident inflation at their minimum values, equal to fifty by scale construction[18] (Fig. 1a).

Small per-feature effect sizes and zero-inflated questionnaire responses place tremendous challenges to investigating the neural etiology of psychopathology in population samples of youths. Because psychopathology is manifested in a minority of individuals and may gradually emerge during adolescence, domain scores from psychiatric questionnaires (e.g., attention problems or depressive withdrawal symptoms) are typically zero-inflated in these samples. In these situations, researchers typically resort to general summaries of the aggregated scores, such as total problems, instead of domain scores, in an attempt to circumvent the issue of zero-inflated, skewed outcomes. However, association patterns with the general summary score from the CBCL are global without regional specificity[19,20], hence providing limited insight into the neural mechanisms of domain symptom profiles. Prior, small-scale studies suggest that domain scores might have more regional specificity[6,21,22], although there are inconsistencies in reported regions, as some studies have implicated ventrolateral prefrontal cortex in attention problems[6] whereas others have reported involvement of the dorsal anterior cingulate cortex[21,22]. A recent analysis from a large-scale study found no evidence for regional differences between adolescents with and without attention deficits and hyperactivities (ADHD) diagnoses in cortical thickness, surface area, and volumes[23]. Those inconsistencies complicate our understanding on the psychopathology among adolescents.

We developed a two-pronged approach to solve the challenge of understanding the psychopathology among youth (Fig. 1b). First, instead of examining one brain region at a time, we can estimate the theoretical bound of the variance explained given all imaging features from a given MRI modality. A similar concept has been explored before[24,25]. Yet, these methods do not provide inferences about candidate causal brain regions[25]. Second, we need a modeling approach that can handle zero-inflated outcomes. Existing methodologies are built either for continuous normally-distributed or binary outcomes[25]. Mis-specifying semi-continuous outcomes as being normally distributed leads to incorrect estimation of their variances, generally leading to a downward bias in the estimation on total variance explained[26]. No exisitng analytical techniques are specifically developed to model the relationship between semi-continuous traits and whole-brain imaging features.

Here, we present our newly developed Bayesian model, the zero-inflated variance estimator (ZIV; https://github.com/junting-ren/ZIV). Figure 1 illustrates the design of ZIV. Given a set of whole-brain imaging features in the region-of-interest (ROI) level, ZIV estimates the total variance explained by all features *en masse* (fraction of variance explained [FVE]; Fig. 1c, top section), proportion of non-null effects among all included features (Proportion of non-nulls [PNN]), and identifies the most important features by estimating a probability for each feature (feature non-null probability [FNNP]; Fig. 1c, middle section). Although the PNN estimates show bias, the combined use of FNNP allows for the selection of true causal features with high sensitivity and a low false discovery rate, as demonstrated in simulations. The posterior weights from the ZIV estimator can then can be used for prediction in an independent sample (Prediction Weights; Fig. 1c, bottom section).

ZIV achieves these goals by combining point-and-slab priors with the Tobit model, capturing the global-local signals of multiple imaging features simultaneously from whole-brain images and the zero-inflated outcomes. The practicality of the ZIV estimator is strengthened by flexible implementations, using either a Variational Bayes algorithm for speed or a Monte Carlo Markov Chain (MCMC) algorithm for more accuracy (denoted as ZIVM). We demonstrate the validity and the utility of our method with comprehensive Monte Carlo simulations and empirical applications on the Adolescent Brain Cognitive Development[SM] (ABCD) Study.

## Results
### Monte Carlo simulation results
In the following sections, we present the result for 200 instances of Monte Carlo Simulation with outcome generated by using either independent standard normal features or ABCD study task-functional MRI (tfMRI) image features. For outcomes generated by using independent standard normal features, we fix the true FVE, PNN and number of features constant at 0.5, 0.1 and 400 respectively, while varying the number of observations to assess the validity and consistency of our model's estimations. For the simulated independent features, we examine the impact of zero-inflation on model performance by generating outcomes with (truncated) and without zero-inflation (linear). For the zero-inflated outcomes generated by using ABCD study tfMRI image features, we utilize the real tfMRI images feature matrix encompassing 8893 subjects and 885 features, and vary the the true FVE and PNN to evaluate model performance across different real-data scenarios characterized by highly correlated features. We benchmark the FVE estimates from ZIV and ZIVM models against those from GCTA[27], a liability-based model initially proposed for genetic studies and subsequently adapted for neuroimaging data analysis[25,28]. GCTA expects the lower

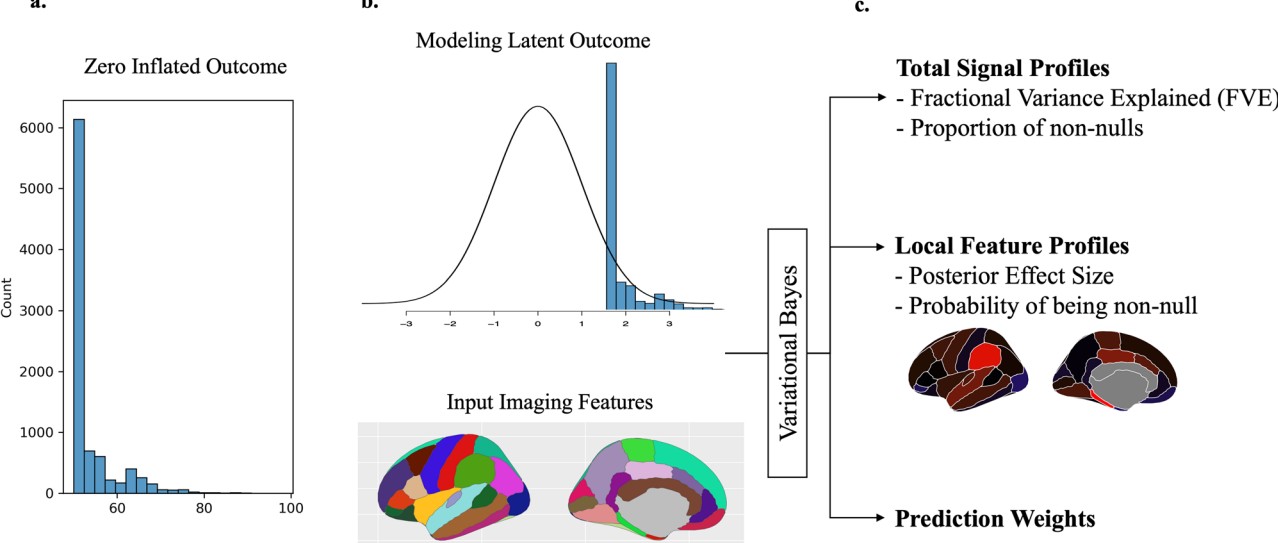

**Fig. 1 | ZIV schematic. a**. The histogram of a zero-inflated outcome. The data is highly concentrated at 0 with a long right tail. **b** ZIV assumes a latent outcome and the input imaging features, such as region-of-interest measures, have a linear relationship. **c** Through variational Bayes algorithm, ZIV estimates both total signal profiles and the local feature characteristics simultaneously. The resulting posterior weights can be used for predictions and feature selection.

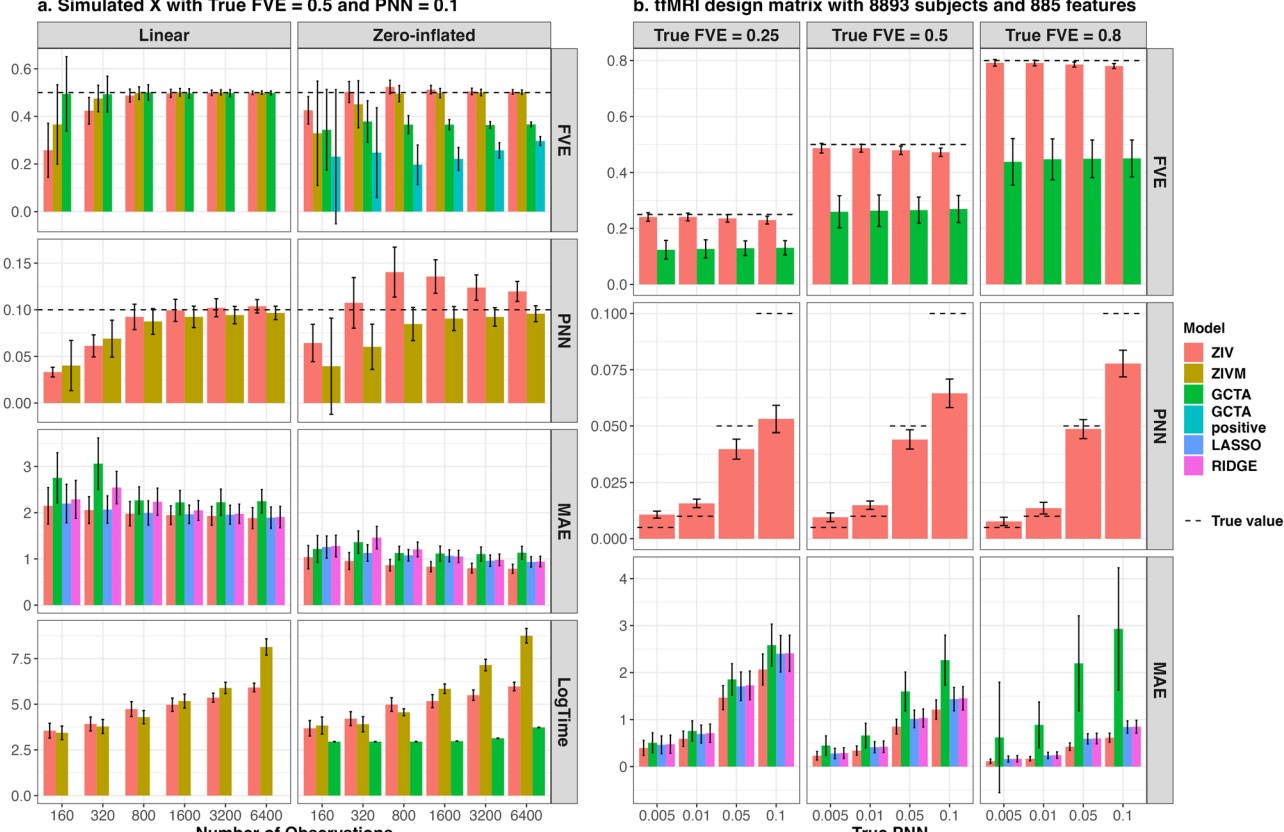

**Fig. 2 | Monte Carlo Simulation Result of point estimation for FVE, PNN and MAE under different types of outcomes, design matrices, number of sample size and true FVE/PNN values.** The dashed lines display the true values. Error bars extend to one standard deviation both above and below the mean of the point estimates. For each simulation setup, a total of 200 instances are conducted to determine the mean and standard deviation of the point estimates. Panel **a** is simulated using independent standard normal features with true FVE, PNN and number of features fixed at 0.5, 0.1 and 400, where the distributions of the point estimates are shown as a function of sample size and outcome characteristics. Panel **b** is simulated using ABCD tfMRI image features with 8893 subjects and 885 features. While the sample size and the number of features are fixed, the distributions of the point estimates are shown as a function of true PNN and FVE. FVE fraction of variance explained, PNN proportion of non-null, MAE mean absolute error, Log-Time natural log of computational time in seconds, ZIVM ZIV model estimated using MCMC algorithm.

triangular part of the relatedness matrices as the input for estimating FVE. We calculated the relatedness matrix given the correlations across input imaging features[25,28], and then call GCTA to perform the estimation given the imaging-based relatedness matrix. In addition to FVE, GCTA also output BLUP for the prediction. We also implemented Ridge regression and Lasso, using scipy, to compare the prediction accuracy across models, utilizing MAE as the metric.

## Parameter point estimation, prediction accuracy and computational time

For FVE estimation using simulated independent standard normal features, as illustrated in the top row of Fig. 2a, the FVE estimates from both ZIV and ZIVM models converge to the true value of 0.50 as the sample size increases, for both linear and zero-inflated outcomes. GCTA showed a similar trend in linear outcome, but significantly underestimates the FVE irrespective of whether it is trained on all observations or solely on positive ones. Even though the PNN estimation exhibits bias for zero-inflated outcomes, both the ZIV and ZIVM estimation of PNN displayed a trend of converging to the true value as sample size increases as shown in the second row, whereas the GCTA does not provide estimation for PNN. The third row of Fig. 2a showcases the superior prediction accuracy of the ZIV model over GTCA, LASSO, and Ridge regression under the assumption of sparsity, regardless of outcome type. The fourth row illustrates the logarithm of computational time in seconds for ZIV, ZIVM, and GCTA models, highlighting a linear increase in computational time with the number of observations for ZIV

and GCTA, in contrast to the polynomial time increase observed with ZIVM. Given the analogous performance of ZIV and ZIVM for scenarios where the observation count exceeds the feature count, only the ZIV model was implemented and compared with GCTA in simulations based on ABCD tfMRI features.

Figure 2b display the results for outcome simulated by highly correlated ABCD tfMRI features. Across varying true FVE and PNN values, ZIV marginally underestimates FVE by approximately 0.02, while GCTA drastically underestimates it by around 0.10, 0.15 and 0.30 when the true FVE equals 0.25, 0.5, 0.8, respectively. ZIV's PNN estimates show a tendency to overestimate at lower true PNN values and underestimate at higher ones. In terms of prediction performance, ZIV models surpass GCTA, LASSO, and Ridge predictions, with average improvements of 49.7%, 19.8%, and 21.2% over GCTA, LASSO, and Ridge, respectively, in MAE across all scenarios.

## Credible interval coverage rate and range

For uncertainty quantification, both ZIV and ZIVM provide credible intervals (CI) for FVE and PNN estimates. As illustrated in the top figure of Fig. 3, the coverage rate of CIs for FVE provided by ZIV and ZIVM aligns with the anticipated 95% nominal level for linear outcomes. However, for zero-inflated outcomes, while ZIVM's coverage rate converges to the 95% nominal threshold, ZIV's coverage rate surpasses it. Regarding the PNN coverage rate, both ZIV and ZIVM models exceed the nominal level when the sample size ($n$) exceeds the number of predictors ($p$). The bottom figure

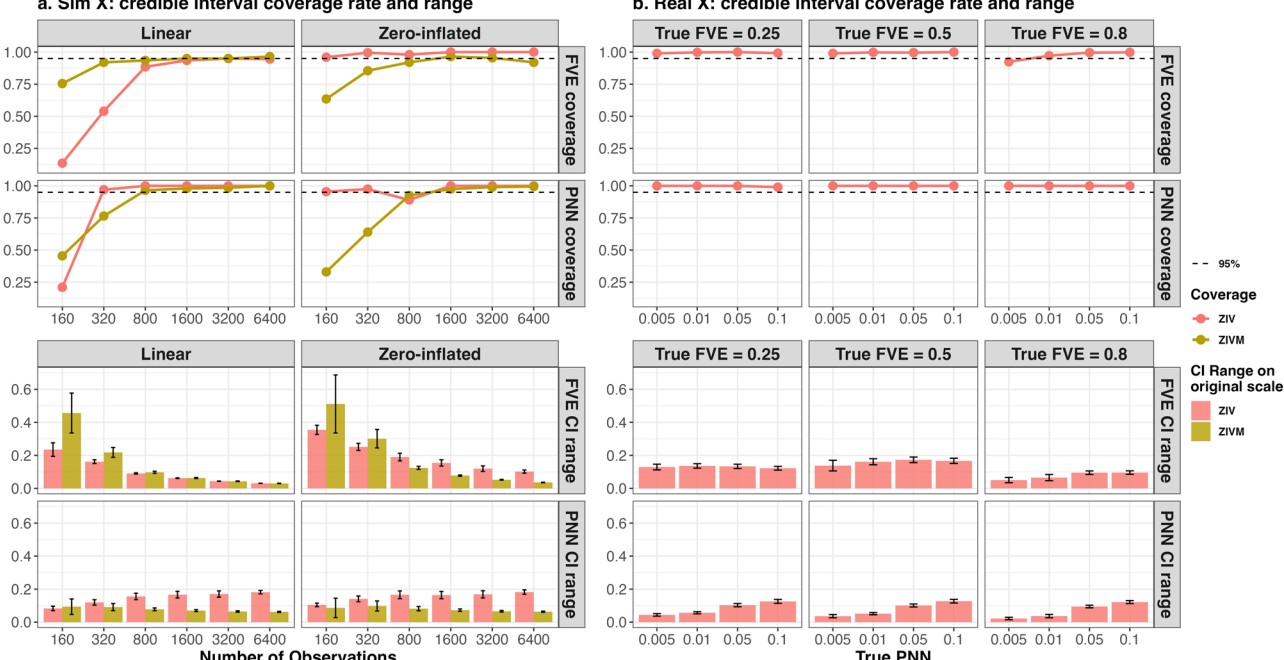

**Fig. 3 | Monte Carlo Simulation Result of CI coverage rate and range for FVE and PNN under different types of outcomes, design matrices, number of sample size and true FVE/PNN values.** The dashed lines display the nominal coverage rate of 95%. Error bars extend to one standard deviation both above and below the mean of the range. For each simulation setup, a total of 200 instances are conducted to determine the mean and standard deviation of the coverage rate and range. Panel **a** is simulated using independent standard normal features with true FVE, PNN and number of features fixed at 0.5, 0.1 and 400. Panel **b** is simulated using ABCD tfMRI image features with 8893 subjects and 885 features. CI credible interval.

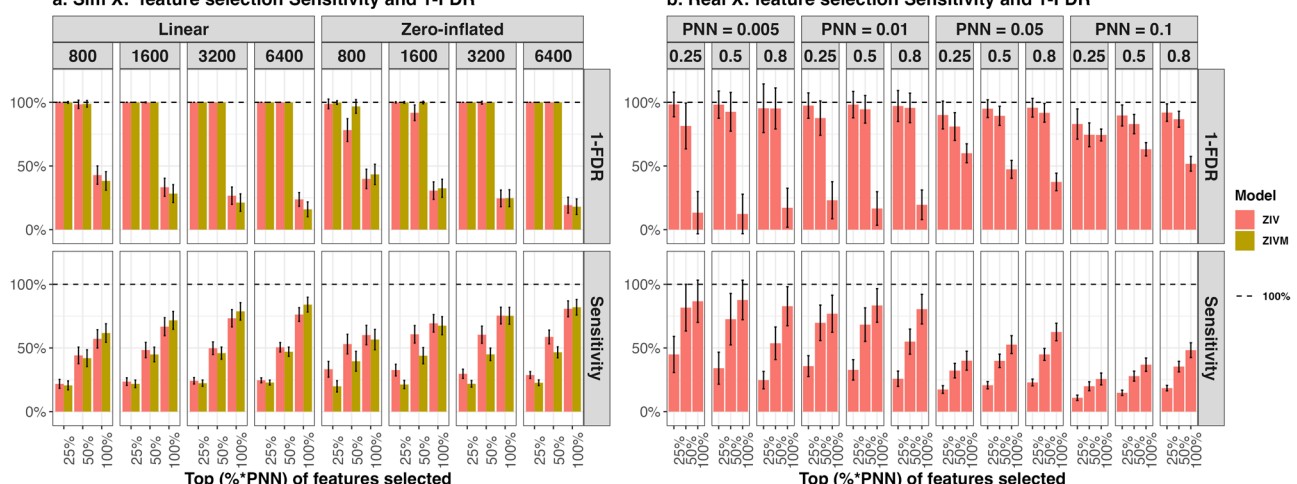

**Fig. 4 | Monte Carlo Simulation Result of feature selection sensitivity and false discovery rate using estimated global PNN and local FNNP under different types of outcomes, design matrices, number of sample size and true FVE/PNN values.** The dashed lines display the position of 100% on y-axis. Error bars extend to one standard deviation both above and below the mean. For each simulation setup, a total of 200 instances are conducted to determine the mean and standard deviation of the sensitivity and false discovery rate. Panel **a** is simulated using independent standard normal features with true FVE, PNN and number of features fixed at 0.5, 0.1 and 400. Panel **b** is simulated using ABCD tfMRI image features with 8893 subjects and 885 features. FDR: false discovery rate; PNN: proportion of non-null; FNNP: individual feature non-null probability.

of Fig. 3 displays the range of the CIs. The CI ranges of ZIV are consistently larger than that of ZIVM when $n > p$. The CI ranges for FVE decreases as sample size increases for both models, whereas the CI ranges for PNN stays relatively constant. Although the coverage rate is higher than nominal level for some setup, the relatively small range of the CI, comparing to the original scale of the estimates, still offers meaningful information of the point estimates.

When utilizing ABCD tfMRI features to generate outcomes, the coverage rates of the CI consistently exceed the nominal level for different true FVE and PNN, indicating that the CIs are wider than expected, which reduces the power of the CIs. Nevertheless, the CI sustains coverage rates above the nominal level across different setups and provides valid inference within a relatively narrow range. The range of the CI for FVE decreases as the true FVE increases, and the range of the CI for PNN increases as the true

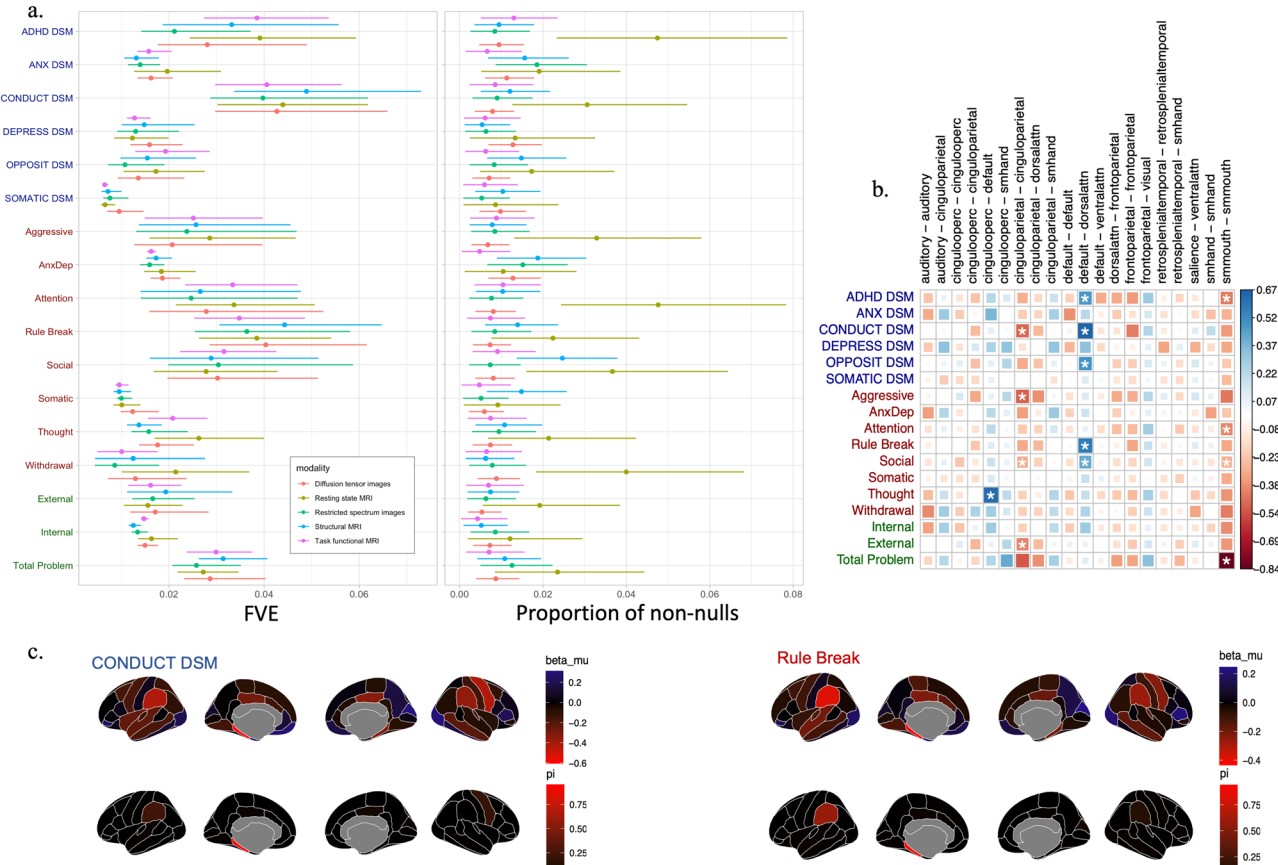

**Fig. 5 | Brain and CBCL behavior associations in ABCD. a** The global signal architectures of CBCL across imaging modalities. **b** Posterior estimates of the local features of rsfMRI. The coloring and size of the square in a cell represent the posterior effect size of each measure pair while the asterisk marks indicate the probability of being non-null exceeding 50%. **c** Posterior estimates of the local features of sMRI for two CBCL scales that have highest FVE. The posterior effect sizes are illustrated on the upper row and the posterior probabilities are illustrated on the lower row.

PNN increases. This pattern underscores the ZIV's perception of increasing uncertainty in scenarios characterized by diminished information availability (low FVE) or heightened noise presence (attributable to elevated PNN and feature correlation).

**Feature selection sensitivity and false discovery rate**

The ZIV model estimates the proportion of all the features being non-null, denoted as PNN ($\pi$), and for each individual feature $j$, estimates the non-null probability, denoted as FNNP ($\pi_{\beta_j}$). At the global level, the estimation of PPN $\pi$ implies that the total number of non-null features should not exceed $p \times \pi$ where $p$ is the total number of features. Therefore, in the Monte Carlo experiments below, we identify the top $k$ features with highest FNNP $\pi_{\beta_j}$ as non-null, where $k$ is defined as $t \times \hat{\pi} \times p$. We set $t = (100\%, 50\%, 25\%)$ to demonstrate the trade off between sensitivity (out of all true non-null features, the percentage selected) and false discovery rate (FDR, out of all selected features, the percentage being true non-null). In Fig. 4a, which illustrates outcomes derived from simulations using independent features, no feature was falsely identified as non-null when the percentage $t$ was lower than 50%. Additionally, the sensitivity was maintained around 50% across various simulation settings. This indicates that the features captured by the model all are true non-null and out of all true non-null features, 50% of them are selected by our model. In Fig. 4b for outcomes generated using ABCD tfMRI image features, a similar trend can be observed albeit with a higher FDR and lower sensitivity compared to the independent feature simulations. Notably, in the simulated scenario that closely mirrors real data characteristics (true FVE=0.25 and PNN=0.005), setting $t$ to 25% results in an FDR near zero while still capturing about 50% of the true non-null features.

This analysis underscores the model's efficacy in distinguishing between null and non-null features under varying simulation conditions, illustrating its potential applicability in real-world data analysis.

**ABCD study results**

We used ZIV to infer the signal architecture of the CBCL subscales, one imaging modality at a time. Results are summarized in Fig. 5a. Across CBCL scales, the estimated FVE ranges from 0.6% to 4.9%. Imaging modalities have a similar range of FVE's given the same CBCL subscale, with the highest magnitude of FVE in the DSM-oriented Conduct subscale and the Rule Break syndrome subscale. The proportion of non-nulls also has similar range across all imaging modalities except rsfMRI. Across CBCL subscales, rsfMRI has the highest proportion of non-null compared to other imaging modalities. It also has the largest model uncertainties in PNN compared to others.

The estimated PNN are all less than 5% of included features. Posterior estimates thus indicate brain-behavior signals are concentrated on smaller subsets of ROIs with an overall background of weaker effects (Fig. 5b and c). Between-network connectivity between the default mode and dorsal lateral attention networks is associated with CBCL scales in three DSM-oriented scales (ADHD, Conduct, and Opposition) and two syndrome scales (Rule break and Social). Within cingulo-parietal network connectivity, on the other hand, is found to be most closely related to externalizing behaviors. The somatosensory network has strong associations with all CBCL sub-scales, although most prominently in the ADHD, Attention, Social, and Total Problem subscales (Fig. 5b). We do note that the PNN may be underestimated or overestimated due to the biasedness shown in simulation.

Mirroring results from rsfMRI, local signals from the associations with sMRI measures were more evident in the cingulate, parietal, and somatosensory regions, as shown in the Fig. 5c. The volume of the parahippocampus was inferred to have the highest probability of being non-null compared to all other regions.

## Discussion

We demonstrated that the ZIV model can efficiently infer the total global signal while simultaneously localizing the driving features, given zero-inflated outcomes and high-dimensional imaging predictive features. In simulations using realistic high-dimensional features from task fMRI, ZIV out-performed other tools in estimating the true values of FVE and proportion of non-null signal, as well as in predictive accuracy on independent testing sets. By adopting variational inference, ZIV provides posterior estimates within minutes using a modern laptop computer. ZIV is hence useful and practical for application to large-scale brain-behavior analyses of many existing datasets.

Analyzing zero-inflated outcomes without considering the violation of normality assumption can lead to serious bias. As showcased in our simulations, FVE estimated by methods like GCTA are severely downward biased. ZIV formally models the zero-inflated outcome as a result of truncation of a partially observed latent variable, hence more accurately capturing the global signal architecture. With correct model specification, ZIV also improves prediction accuracy, performing better than GCTA, LASSO, and Ridge regression in all settings.

Our empirical findings of applying ZIV to the ABCD cohort indicate that, regardless of imaging modality, relevant brain features are most consistently detected for externalizing symptoms. Conduct problems, rule breaking behaviors, and total problem scales exhibited the highest FVE across measures. It is perhaps unsurprising that externalizing symptoms were most strongly linked with brain morphology and function, with ADHD and conduct problems forming some of the most prevalent mental disorders in early adolescence[29]. Indeed, recent analysis within the ABCD sample found these behaviors to be most strongly predicted by genetics[30]. Taken together these results indicate that externalizing symptom assessments may exhibit more variability in this young adolescent sample leading to a greater ability to detect associations.

A recent high impact paper used ABCD data to argue that the strength of brain-behavior associations were much smaller than previously thought[8]. In this work researchers presented cross-validation predictions claiming that rsfMRI features explain around 1 percent of variance of the CBCL Total Problem while much less for the CBCL internalizing and externalizing measures. We find this previous work likely underestimated proportion of variance explained, in part due to a mispecified model assuming normality of variables. the current work tackles this problem directly and in so doing provides a more comprehensive picture of associations between brain and adolescent mental health.

In particular, our estimates of the proportion of non-null effects indicate that, for the item-level behavioral measures among youth, brain-behavior signals are not ubiquitous across brain regions. This is in contrast with reports that focus on more complex and normally distributed outcomes, such as intelligence scores[31]. This sparseness also violates assumptions used in methods such as GCTA[25,27,28], rendering them inappropriate to model such effects. The sparseness of the non-null effects across brain regions enables analyses to partition out neural circuitry related to components of behavior - such as those captured by item-level measures. Here we have showcased that ZIV is well suited for this purpose since many of the item-level behavioral measures are zero-inflated and heavily skewed.

Our analyses demonstrated that default mode network connectivity with the dorsal lateral attention network has consistent associations with ADHD, Conduct, Opposition, Rule breaking, and Social scales. The mis-engagement of the default mode network with the attention network has been posited as a key driver for attention issues among youth[32]. Concordant with the rsfMRI results, our analysis on the volumetric measures from sMRI show that the reduced volumes in parietal, cingulate, and parahippocampal regions are consistently associated with Conduct and Rule breaking scales. These results suggest that circuitry linking these regions is more salient to externally-manifested behavior, rather than attention alone.

Surprisingly, subcortical structures, such as hippocampus, do not exhibit stronger signals than other brain regions in our analyses. The posterior effect sizes are concentrated on the cortical regions. While such observations are consistent with prior reports based on youth samples[6,21,22], it is in sharp contrast to the results from the ENIGMA consortium[33]. The meta-analyses on the case-control designed studies show most of the consistent differences between patients with psychiatric disorders and healthy controls are located at hippocampus[33]. It is possible what we capture here, is the neural substrate underlying normal variations of the psychopathologies among youth or the early precursor of a disease process. Our application of ZIV is the first step toward the understanding of the etiology of psychiatric disorders.

Our model addresses the zero-inflated nature of the data through modeling a single latent variable, whereas traditional two-part models treat the zero as true values and separately describes the probability of the outcome being positive and the magnitude of positive values[12,34,35]. The two-part models offer flexibility by not presuming data below the detection threshold (zero) as unobserved. However, for brain imaging applications, the ZIV (single model approach) demonstrates greater suitability. First, the latent model provides enhanced interpretability, estimating FVE with easily comprehensible coefficient effects. In contrast, the two-part model does not directly estimate FVE and requires interpreting two distinct sets of coefficients. Second, when considering behavioral outcomes, it is logical to assume the behavior remains unobserved until the latent variable surpasses a certain threshold.

In addition, our Bayesian model inherently incorporates multiple testing correction by estimating the PPN $\pi$ in the model[36], which indicates that the proportion of true causal features is likely to be less than $\pi$. Consider the simulated scenario where the true PNN and FVE is set at 0.05 and 0.8 respectively, as shown in Fig. 2b. Our model's inferred PNN is 0.049, which suggests that the proportion of features with a positive association is below 5%. Consequently, when selecting significant features, we would identify fewer than 5% of them as having a high probability of being non-null, based on each feature's local posterior non-null probability. Figure 4 demonstrates that this approach achieve high sensitivity and low FDR under various setup. It contrasts with the multiple testing strategy used in Genome-Wide Association Studies (GWAS), where each feature is evaluated independently using univariate regression[37]. In GWAS, features are deemed significant based on their individual p-values, without an aggregate estimate of the proportion of non-null features across the entire set.

The ZIV model has been validated with comprehensive simulations. When simulating zero-inflated outcomes with independent features, we observe that point estimates FVE and PNN progressively align with the ground truth as the sample size increases. However, we do note that the PNN still exhibits bias and the simulation setup is rather ideal and needs to be interpreted with caution. For simulations utilizing real tfMRI features, the model slightly underestimates FVE and inconsistently estimates the PNN: overestimating at lower true values and underestimating at higher ones. The FVE underestimation is linked to the zero-inflation censoring, leading to incomplete information, while the PNN inconsistency arises from the high correlation among tfMRI features. These findings underscore the need for further investigation to improve the estimation and inference processes related to PNN in future studies. Because we do not explicitly assign different priors for the input subset and do not include bivariate outcomes, we cannot test the signal overlaps between imaging modalities. It remains to be seen if the converging results we found across modalities in the ABCD data are indeed tagging the same biological signals. Despite these limitations, it is important to highlight that the ZIV model, when compared to existing methodologies such as GCTA, offers superior performance. This is evident in the higher accuracy in FVE and prediction. Moreover, although the estimation of the PNN exhibits bias, the ZIV model effectively utilizes PNN to select important features with high sensitivity and low false discovery rate.

In sum, our development and efficient implementation of the ZIV mdel provides a necessary tool to investigate brain-behavioral relationships zero-inflated, highly-non-normal data. In these cases, ZIV produces unbiased estimates of the global signal from high-dimensional imaging data while providing detail on signal localization. Because of the high prevalence of semi-continuous data in many fields, ZIV could be applied to analyses beyond brain-behavior research.

## Methods

### Model overview

The core of ZIV estimator is a Tobit model with *spike-and-slab* priors imposed on its coefficients. This model bears resemblance to traditional linear regression but is specifically tailored for situations where the dependent variable is zero-inflated by being censored at a certain threshold. ZIV enables the application of standard linear regression methods on uncensored data, while treating observations as censored values, not exact values[38]. The Tobit model is widely used in health outcome research. For instance, it is utilized for self-reported psychometric scales[39], for exploring the relationship between Everyday Cognition scales and structural neuroimaging[40], and for examining the link between memory functions and the 5-HT type 4 receptor[41].

The spike-and-slab prior is a Bayesian approach to variable selection and coefficient estimation, utilizing a mixture of two distributions: a "spike" representing a point mass at zero for irrelevant features and a "slab" representing a continuous distribution for relevant features. Imposing a spike-and-slab prior on coefficients of a linear model aids in variable selection and mitigates overfitting in high-dimensional settings[42]. It has been frequently used in identifying causal variables with various degree of background correlations, such as fine mapping in genetic applications[43]. The spike-and-slab method has also been utilized to improve the interpretation and detection of neural activity in various studies, such as those involving calcium imaging[44], electromagnetic brain mapping[45], and functional MRI[46,47]. Given the goal to identify key neural pathways underlying complex neurobehaviors, e.g. psychopathology among youth, we adopt this strategy in our ZIV implementation.

The observed semi-continuous outcome $z_i$ for subject $i$ is modeled by positing a latent variable $y_i$,

$$z_i = \begin{cases} y_i & , y_i > 0 \\ 0 & , y_i \leq 0 \end{cases} \tag{1}$$

The latent variable and feature pairs $\{(y_i, \boldsymbol{x}_i)\}_{i=1}^{n}$ are assumed to follow a linear relationship:

$$y_i = \beta_0 + \boldsymbol{x}_i^T \boldsymbol{\beta} + \epsilon_i \tag{2}$$

where the error terms $\epsilon_i$ are assumed to be independently and identically distributed as $N(0, \sigma^2)$, $\boldsymbol{x}_i = (x_{i1}, x_{i2}, ..., x_{ip})^T$ and $\boldsymbol{\beta} = (\beta_1, \beta_2, ..., \beta_p)^T$ are both $p$-dimensional column vectors. We also denote the $n$ by $p$ design matrix as $\boldsymbol{X} = (\boldsymbol{x}_1, \boldsymbol{x}_2, ..., \boldsymbol{x}_n)^T$.

### Prior Specification

We model the effects of the features as a mixture of priors from normally distributed non-nulls and point mass nulls (slab and spike prior):

$$\beta_j \sim N(0, \sigma_\beta) \pi + \delta_0(\beta_j)(1 - \pi) \tag{3}$$

where $\delta_0$ denotes a point mass at zero. This formulation implies the following: when $\beta_j = 0$, then $\delta_0(\beta_j) = 1$, leading the density to be the sum of $1 - \pi$ and $\pi$ times the normal density value evaluated at zero. On the other hand, When $\beta_j \neq 0$, $\delta_0(\beta_j) = 0$, and the density simplifies to $\pi$ times the normal density at $\beta_j$. We reparameterize $\beta_j$, $j = 1, ..., p$, as

$$\beta_j = \delta_j \widetilde{\beta}_j \tag{4}$$

where

$$\begin{aligned} \widetilde{\beta}_j &\sim N(0, \sigma_\beta^2) \\ \delta_j &\sim \pi^{\delta_j}(1 - \pi)^{\delta_j} \end{aligned} \tag{5}$$

Reparameterization can simplify the model's structure, making it easier to understand and interpret. By expressing $\beta_j$ as a product of $\delta_j$ and $\widetilde{\beta}_j$, this effectively separates the mixture model into its components. This decomposition not only facilitates a more intuitive understanding of the model but also simplifies the derivation process in subsequent steps.

The specification of priors for other parameters in our model is contingent upon the chosen posterior inference methods:

- For Variational Inference method, we assume the following non-informative prior for the global proportion of non-nulls, $\pi$, and the variance of the error terms, $\sigma^2$, as:

$$\begin{aligned} \pi &\sim \text{Uniform}(0, 1) \\ \sigma^2 &\sim \text{Uniform}(0, +\infty) \end{aligned} \tag{6}$$

We denote by $\theta = (\sigma_\beta, \beta_0)$ the parameters optimized using gradient descent without a variational posterior.

- For Markov Chain Monte Carlo Inference Method, we assume the following non-informative priors:

$$\begin{aligned} \pi &\sim \text{Beta}(0.5, 0.5) \\ \sigma^2 &\sim \text{InverseGamma}(0.1, 0.1) \\ \sigma_\beta^2 &\sim \text{InverseGamma}(1, 1) \\ \beta_0 &\sim \text{Normal}(0, \sigma_\beta^2) \end{aligned} \tag{7}$$

### Posterior inference methods

The objective of posterior inference is to obtain the posterior distribution of the model parameters, given by:

$$\mathbb{P}(\widetilde{\boldsymbol{\beta}}, \boldsymbol{\delta}, \pi, \sigma | \boldsymbol{X}, \boldsymbol{z}) \propto \mathbb{P}(\boldsymbol{z} | \widetilde{\boldsymbol{\beta}}, \boldsymbol{\delta}, \pi, \sigma) \times \mathbb{P}(\widetilde{\boldsymbol{\beta}}, \boldsymbol{\delta}, \pi, \sigma) \tag{8}$$

where $\mathbb{P}(\boldsymbol{z} | \widetilde{\boldsymbol{\beta}}, \boldsymbol{\delta}, \pi, \sigma)$ denotes the data likelihood and $\mathbb{P}(\widetilde{\boldsymbol{\beta}}, \boldsymbol{\delta}, \pi, \sigma)$ denotes the prior distribution.

The posterior distribution encompasses a variety of random variables, including continuous ones like $\widetilde{\boldsymbol{\beta}}$ and discrete ones such as $\boldsymbol{\delta}$. Due to this mix of variable types and high dimensionality of the parameters, deriving a closed-form expression of the distribution for direct computation is impractical.

Therefore, we provide two different estimation algorithms: Markov Variational Inference (VI) and Chain Monte Carlo (MCMC). MCMC, a widely used method, constructs a Markov chain for sampling from the posterior distribution, offering precise estimation[48]. In contrast, VI, a more recent technique, approximates the posterior with simpler parametric distributions[49]. While VI is more efficient in terms of computation and scalability, it tends to offer approximations that are less precise compared to the direct sampling approach of MCMC[49]. Offering two estimation algorithms provides users with the flexibility to choose based on their specific requirements, whether prioritizing higher speed or higher accuracy.

### Variational inference method

We want to approximate the true posterior shown in Equation (8). To achieve this, we use the following variational distributions for

approximation:

$$
\begin{aligned}
q_\phi(\widetilde{\boldsymbol{\beta}}, \boldsymbol{\delta}, \pi, \sigma) &= \prod_{j=1}^{p} N\left(\tilde{\beta}_j | \mu_{\beta_j} \delta_j, \sigma_{\beta_j}^2 \delta_j + \sigma_{\beta_j}^2(1-\delta_j)\right) \\
&\times \prod_{j=1}^{p} \text{Bernoulli}(\delta_j | \pi_{\beta_j}) \\
&\times \text{Beta}(\pi | a_3, a_4) \\
&\times \text{LogNormal}(\sigma^2 | \mu = b_3, \sigma^2 = b_4)
\end{aligned}
\tag{9}
$$

The deliberate decision to maintain independence among these variational distributions is to simplify the estimation algorithm. It's important to note, however, that in reality, the true posterior for the parameters should be correlated: the true posterior of $\widetilde{\boldsymbol{\beta}}, \boldsymbol{\delta}, \pi, \sigma$ is a multivariate distribution, which is intractable. Our approach with the variational distribution is to approximate this true posterior using simpler parametric forms. We introduce variational parameters as $\phi = (a_3, a_4, b_3, b_4, \boldsymbol{\mu}_\beta, \boldsymbol{\sigma}_\beta, \boldsymbol{\pi}_\beta)$, where $\boldsymbol{\mu}_\beta = (\mu_{\beta_1}, \mu_{\beta_2}, ..., \mu_{\beta_p}), \boldsymbol{\sigma}_\beta = (\sigma_{\beta_1}, \sigma_{\beta_2}, ..., \sigma_{\beta_p}), \boldsymbol{\pi}_\beta = (\pi_{\beta_1}, \pi_{\beta_2}, ..., \pi_{\beta_p})$. Once these variational parameters $\phi$ are learned from data, we can sample from $q_\phi$ to approximate the true posterior distribution. For instance, to obtain the posterior distribution of $\delta_j$, we simply sample from Bernoulli($\pi_{\beta_j}$).

To measure the distance between the true posterior and the approximate posterior, we use the Kullback-Leibler divergence (KL-divergence):

$$
\begin{aligned}
D_{KL}(q \parallel p) &= \mathbb{E}_{q_\phi}\left(\log \frac{q_\phi(\widetilde{\boldsymbol{\beta}}, \boldsymbol{\delta}, \pi, \sigma)}{\mathbb{P}(\widetilde{\boldsymbol{\beta}}, \boldsymbol{\delta}, \pi, \sigma | \boldsymbol{X}, \boldsymbol{z})}\right) \\
&= \mathbb{E}_{q_\phi}\left(\log q_\phi - \log \mathbb{P}(\widetilde{\boldsymbol{\beta}}, \boldsymbol{\delta}, \pi, \sigma, \boldsymbol{X}, \boldsymbol{z}) + \log \mathbb{P}(\boldsymbol{X}, \boldsymbol{z})\right) \\
&= \mathbb{E}_{q_\phi}\left(\log q_\phi - \log \mathbb{P}(\widetilde{\boldsymbol{\beta}}, \boldsymbol{\delta}, \pi, \sigma, \boldsymbol{X}, \boldsymbol{z})\right) + \log \mathbb{P}(\boldsymbol{X}, \boldsymbol{z})
\end{aligned}
\tag{10}
$$

Therefore, we have

$$
\log \mathbb{P}(\boldsymbol{X}, \boldsymbol{z}) = D_{KL}(q \parallel p) + \mathbb{E}_{q_\phi}\left(\log \mathbb{P}(\widetilde{\boldsymbol{\beta}}, \boldsymbol{\delta}, \pi, \sigma, \boldsymbol{X}, \boldsymbol{z}) - \log q_\phi\right)
\tag{11}
$$

Since $\log \mathbb{P}(\boldsymbol{X}, \boldsymbol{z})$ is a constant, in order to minimize $D_{KL}(q \parallel p)$, we maximize $\mathbb{E}_{q_\phi}\left(\log \frac{\mathbb{P}(\widetilde{\boldsymbol{\beta}}, \boldsymbol{\delta}, \pi, \sigma, \boldsymbol{X}, \boldsymbol{z})}{q_\phi(\widetilde{\boldsymbol{\beta}}, \boldsymbol{\delta}, \pi, \sigma)}\right)$ which is the Evidence Lower Bound (ELBO). The ELBO can be written in three separate parts:

$$
\text{ELBO} = \mathbb{E}_{q_\phi}\left[\log \mathbb{P}(\boldsymbol{z} | \widetilde{\boldsymbol{\beta}}, \boldsymbol{\delta}, \pi, \sigma, \boldsymbol{X})\right]
\tag{12}
$$

$$
+ \mathbb{E}_{q_\phi}\left[\log \mathbb{P}(\widetilde{\boldsymbol{\beta}}, \boldsymbol{\delta}, \pi, \sigma, \boldsymbol{X})\right]
\tag{13}
$$

$$
+ \mathbb{E}_{q_\phi}\left[-\log q_\phi(\widetilde{\boldsymbol{\beta}}, \boldsymbol{\delta}, \pi, \sigma, \boldsymbol{X})\right]
\tag{14}
$$

Equation (12) is the expectation of the data likelihood over the variational distributions, Equation (13) is the expectation of the prior likelihood and Equation (14) is the entropy of the variational approximation distribution. We use Adam[50] for our stochastic gradient descend algorithm to minimize the negative of the ELBO. The details of the posterior inference and optimization methodologies are shown in the Supplementary Materials Note 1.

### Markov Chain Monte Carlo inference method
MCMC methods constitute a group of algorithms designed for sampling from complex probability distributions that are difficult to compute directly. In our context, the target is the posterior distribution as detailed in Equation

(8). The implemented algorithm is a hybrid approach that integrates both Gibbs and Hamiltonian Monte Carlo (HMC) sampling techniques. While Gibbs sampling can be slow to converge to the target distribution in high-dimensional spaces due to inefficient random walk patterns[51], HMC addresses this issue through a novel auxiliary variable approach. This approach effectively transforms the challenge of sampling from a target distribution into simulating Hamiltonian dynamics[52,53]. However, HMC is designed for continuous model parameters.

Therefore, we employ the Gibbs algorithm for sampling the discrete parameters $\boldsymbol{\delta}$, and HMC for the efficient sampling of continuous parameters ($\widetilde{\boldsymbol{\beta}}, \pi$, and $\sigma$). After obtaining posterior sampling of $\boldsymbol{\delta}$, similar estimation of the $\boldsymbol{\pi}_\beta$ as in the VI algorithm can be calculated simply by taking the mean of $\boldsymbol{\delta}$ across each feature. The algorithm's framework is presented in Algorithm 1, with detailed steps provided in the Supplementary Note 2.

---

**Algorithm 1** Mixed HMC and Gibbs MCMC

---
**Require:** $\tilde{\beta}^{(0)}, \delta^{(0)}, \pi^{(0)}, \sigma^{(0)}, \boldsymbol{X}, \boldsymbol{z}, \boldsymbol{R} = \{\}$
 **for** $m = 1$ to $M$ **do** ▷ Warm Up, non-stationary phase
 $\tilde{\beta}^{(m)}, \pi^{(m)}, \sigma^{(m)} \leftarrow$ HMC sampling condition on $\delta^{(m-1)}, \boldsymbol{X}, \boldsymbol{z}$
 $\delta^{(m)} \leftarrow$ Gibbs Sampling condition on $\tilde{\beta}^{(m)}, \pi^{(m)}, \sigma^{(m)}, \boldsymbol{X}, \boldsymbol{z}$.
 **end for**
 **for** $c = M + 1$ to $C$ **do** ▷ Stationary phase, save the samples
 $\tilde{\beta}^{(c)}, \pi^{(c)}, \sigma^{(c)} \leftarrow$ HMC sampling condition on $\delta^{(c-1)}, \boldsymbol{X}, \boldsymbol{z}$
 $\delta^{(c)} \leftarrow$ Gibbs Sampling condition on $\tilde{\beta}^{(c)}, \pi^{(c)}, \sigma^{(c)}, \boldsymbol{X}, \boldsymbol{z}$.
 Append $\tilde{\beta}^{(c)}, \pi^{(c)}, \sigma^{(c)}, \delta^{(c)}$ to $\boldsymbol{R}$
 **end for**
 **return** $\boldsymbol{R}$

---

### Inferring the Global Signal Architecture of Brain-Behavior Relationships
We use FVE and PNN as the two key metrics to characterize the global signal architecture of brain-behavior associations.

To estimate the FVE on the latent scale, we use the sampled values of latent linear effects in the posterior draw of the parameters. By definition, the variance captured by the latent linear outcome at sample cycle $c$ is given by

$$
\text{var}\left[\widetilde{r}_i^{(c)} | \boldsymbol{x}_i\right],
\tag{15}
$$

where $\widetilde{r}_i^{(c)}$ is the latent linear outcome value for individual $i$ at the current cycle $c$

$$
\widetilde{r}_i^{(c)} = \sum_{j=1}^{p} x_{ij} \tilde{\beta}_j^{(c)} \delta_j^{(c)}.
\tag{16}
$$

Then, the estimated variance in the current cycle $c$ is

$$
\hat{\text{var}}\left[\widetilde{r}_i^{(c)} | \boldsymbol{x}_i\right] = \frac{\sum_i (\widetilde{r}_i^{(c)})^2}{N} - \left(\frac{\sum_i \widetilde{r}_i^{(c)}}{N}\right)^2.
\tag{17}
$$

The estimated FVE on the latent scale is at cycle $c$ is

$$
FVE^{(c)} = \frac{\hat{\text{var}}\left(\widetilde{r}_i^{(c)}\right)}{\hat{\text{var}}\left(\widetilde{r}_i^{(c)}\right) + (\sigma^{(c)})^2}
\tag{18}
$$

where $(\sigma^{(c)})^2$ is the estimate for the noise variance $\sigma^2$ at cycle $c$. We can obtain the posterior distribution for $FVE$ as we repeat the sampling cycle from the posterior distribution for $\beta_j$, $\delta_j$ and $\sigma^2$.

Inferences for PNN can be accomplished directly using the approximated posterior distribution:

$$\pi \sim \text{Beta}(\pi | a_3, a_4). \quad (19)$$

This Bayesian model inherently incorporates multiple testing correction by estimating the PPN $\pi$ in the model, which indicates that the proportion of true causal features is likely to be less than $\pi$. For a more comprehensive discussion and details on this aspect, refer to the Discussion.

## Localized feature inference and selection

For localized feature selection, the model produces posterior distributions $\delta_j \sim \text{Bernoulli}(\pi_{\beta_j})$ that indicates the probability of non-null for each feature $j$. Customized rules based on $\pi_{\beta_j}$ can be utilized to identify likely non-null features. On the global level, the estimation of PPN $\pi$ suggests that the total number of non-null features should be less than $p \times \pi$, which in turn penalizes the sum of all $\delta_j$ should not exceed $p \times \pi$. Therefore, in the Monte Carlo experiments below, we pick the top $k$ highest $\pi_{\beta_j}$ as non-null associations, where $k = t * \hat{\pi} * p$ and $\hat{\pi}$ is the mean of PNN $\pi$ posterior distribution. We set $t = (1, 0.75, 0.5, 0.25)$ to showcase the trade off between false discovery rate and sensitivity.

## Prediction

In our Bayesian model, we can obtain a posterior distribution for $z_i$ through sampling from the posterior distribution multiple times:

$$y_i^{(c)} = \beta_0^{(c)} + \boldsymbol{x}_i^T \boldsymbol{\beta}^{(c)} \quad (20)$$

Then,

$$z_i^{(c)} = \begin{cases} y_i^{(c)} & , y_i^{(c)} > 0 \\ 0 & , y_i^{(c)} \le 0 \end{cases} \quad (21)$$

where $\beta_0^{(c)}$ and $\boldsymbol{\beta}^{(c)}$ are the sampled parameters from their corresponding posterior distribution at cycle $c$.

In order to obtain a point prediction, we directly take the mean of the coefficient posterior distributions and product sum with the corresponding features:

$$\hat{y}_i = \mathbb{E}[\beta_0] + \boldsymbol{x}_i^T \mathbb{E}[\boldsymbol{\beta}] \quad (22)$$

$$\hat{z}_i = \begin{cases} \hat{y}_i & , \hat{y}_i > 0 \\ 0 & , \hat{y}_i \le 0 \end{cases} \quad (23)$$

This is equivalent to sample $z_i$ from the posterior and take the mean of the sampled $z_i$, due to the linearity of the expectations.

## Monte Carlo experiments

We performed Monte Carlo experiments to evaluate the performance of the ZIV model. In order to simulate synthetic data that closely resemble real data, we used actual task fMRI data for our imaging features while varying true model parameters to generate synthesized semi-continuous outcomes. The task fMRI data were sampled from the ABCD 4.0 Data Release. We randomly assigned 0.5%, 1%, 5% or 10% of the features to have non-null effects (i.e., PNN in 0.005, 0.010, 0.050, and 0.100, respectively). The non-null effects were generated from a normal distribution with mean 0 and standard deviation of 0.1. The latent outcome was set equal to the linear combination of the features multiplied by the corresponding coefficients adding normally-distributed noise. The observed outcome was truncated at 0 whenever the latent outcome was negative. Because that the mean of the latent outcome is 0, there are around 50% zeros in the outcomes for each simulated instances. The variance of the random Gaussian noise was determined by the empirical variance of the latent mean (linear combination

of the features multiplied by the corresponding coefficients) and the pre-defined FVE parameter for simulation. We simulated data that have 0.05, 0.25, 0.5 or 0.8 fraction of variance explained. This resulted in a total of 16 different simulation scenarios (4 different percent non-null effects times 4 different pre-defined FVE's). The total sample size and the number of features for the ABCD data simulation are 8893 and 885, respectively. In addition to using real tfMRI data, we generated both linear and zero-inflated outcome under the same setup except that the features are simulated from independent standard normal distribution. We fixed the true FVE, PNN and number of features to be 0.5, 0.1 and 400 respectively, while varying the number of observations to investigate the model consistency. For each simulation setup, we generated 200 instances and aggregated over the estimates of the instances for final result.

First, we examined the performance of ZIV in inferring the global signal architectures, i.e., how well the FVE and PPN covered the true values. For comparison, we also implemented a liability-based linear mixed effects model[25,27,28]. Estimation of morphometricity from liability-based linear mixed effects model is one of the few existing high-dimensional imaging algorithms that focuses on characterizing the global signal[25]. The liability based estimates were first proposed and implemented in the software, called GCTA[27], for human genetic studies and then applied to neuroimaging data[25,28]. In contrast to ZIV, the GCTA model estimates the FVE assuming regression coefficients are normally distributed and hence the signal architecture is ubiquitously non-null across all input features[27,28].

Second, we investigated if ZIV improved the predictive performance by accounting for the zero-inflated distribution and non-null probabilities. We compared the predictive performance of ZIV with three other prediction models: ridge regression, LASSO, and the best linear unbiased predictor (BLUP) from the GCTA model. To evaluate the prediction performance of the models, we randomly split the data into 80% training and 20% testing sets for each outcome-modality pair. The performance metric we used was the mean absolute error (MAE). Compared to mean-squared error (MSE), MAE is less sensitive to outliers that are abundant in non-Gaussian data and hence more appropriate for semi-continuous, highly skewed data. In addition, MAE is more interpretable than MSE because it is assessed in the same units as the data, while the MSE is in squared units. Finally, we compared the computational speed and accuracy between variational inference and the traditional Markov chain Monte Carlo (MCMC) algorithm.

For a fair comparison of prediction accuracy with linear predictors such as Lasso, Ridge and GCTA, we truncate their predictions to 0 when the values are below zero. This ensures that the linear models takes into account the zero-inflated nature of the outcomes.

## Empirical application

We investigated the relationship between the psychopathology and Region of Interest (ROI) brain measurements, one imaging modality at a time, using data from the ABCD Study. The average connectivities among resting state network modules are also included in the analyses, as they are provided by the ABCD data release. The ABCD Study is the largest investigation of neurodevelopment in the United States. N=11,880 youth aged 9-11 at baseline were recruited from 21 different sites around the country; they are currently undergoing annual in-person evaluations for over a decade by the end of the study[54]. Data are released publicly on an annual basis via the National Institute of Mental Health Data Archive (NDA, https://data-archive.nimh.nih.gov/abcd). More details about the study can be found at https://abcdstudy.org/. In this application, we used data from the ABCD 4.0 National Data Archive release (NDAR DOI:10.15154/1523041).

## Multimodal imaging measures

Neuroimaging data were consolidated across all 21 data collection sites and processed by the ABCD Data Analysis Informatics and Resource Center and the ABCD Image Acquisition Workgroup[55]. Data were then obtained at the region of interest (ROI) level for the five MRI modalities available in the

ABCD data release: 1) structural T1 MRI (sMRI), which measures cortical and subcortical morphometry; 2) diffusion tensor images (DTI), which are sensitive to the fiber structures of human brain; 3) restricted spectrum images (RSI), which summarize the properties of tissue compartments; 4) task functional MRI (task fMRI), consisting of event-related contrasts capturing change in the fMRI signal in response to stimuli[56]; and 5) resting state functional MRI (rsMRI), consisting of connectivity measures across Gordon parcellations and subcortical regions, partitioned into modular networks[57]. All included imaging variables and their corresponding categories can be found in the Supplementary Data 5.

The number of features per modality were as follows: 1) $p = 1186$ measures from sMRI; 2) $p = 2376$ measures from DTI; 3) $p = 1140$ from RSI; 4) $p = 885$ from the three fMRI tasks; and 5) $p = 416$ from rsMRI. For all imaging modalities except rsMRI, ROIs were restricted to the Desikan cortical atlas[58]. As described above, rsMRI was based on a modular network partition. Both DTI and RSI features included metrics from segmented major fiber bundles, in addition to the cortical and subcortical ROIs. Casey et al. (2018)[56] offers in-depth information about the imaging acquisition, processing procedures, and quality control metrics in the the ABCD imaging data. All features were standardized to have zero mean and unit standard deviation before entering them into the ZIV models to improve numerical stability and enhance convergence speed for gradient descent optimization.

### Child behavior checklist scores
The Child Behavior Checklist (CBCL) is a tool widely utilized for evaluating an extensive range of emotional and behavioral problems in children[59,60]. It uses a scoring system where responses are labeled as 0 (not applicable), 1 (partially or occasionally applicable), or 2 (completely or frequently applicable). The CBCL is comprised of 113 items that measure aspects of the child's behavior across the past six months. The CBCL provides a total score, along with scores on eight syndrome subscales and six subscales oriented around the Diagnostic and Statistical Manual of Mental Disorders (DSM). The eight syndrome subscales include: (1) anxiety/depression; (2) social withdrawal/depression; (3) somatic complaints; (4) issues with social interaction; (5) thought disturbances; (6) attention issues; (7) rule breaking; and (8) aggressive behavior. Subscales from these eight syndromes can be further summarized into three problem scales: internalizing problems (comprising anxiety/depression, social withdrawal/depression, physical complaints), externalizing problems (comprising violations of rules, hostile behavior), and total problems. The six DSM-oriented scales encompass: (1) depressive disorders; (2) anxiety disorders; (3) somatic disorders; (4) attention-deficit/hyperactivity disorders; (5) oppositional defiant disorders; and (6) conduct disorders. Scores obtained from the CBCL are usually highly right skewed[59]. In particular, the t-standardized score, a preferred scoring that aims to reduce over-interpretation, exacerbates the violation of normal assumption by left truncation of the raw score at 50, leading to inflation at this minimum value[60]. Here, we focus on the t-standardized scores of all eight syndrome scales, six DSM-oriented scales, and three summary problem scales, investigating how brain features associate with these semi-continuous scores.

### Participant inclusion
For each outcome (CBCL scores) and image modality pair, we excluded participants who were missing outcome observations or lacked the corresponding modality ROI measurements. We randomly sampled one member from each family if there were multiple siblings within a family. Within each participant, a single observation was randomly sampled if that participant had multiple MRI assessments across visits. The number of observations included for each image modality for all outcomes is shown in Supplementary Table 1. In all ZIV models we adjusted for race, age, MRI scanner serial info and software versions, and sex at birth as potential confounders. Variance due to these covariates is thus not included in the calculation of FVE for brain imaging features.

### Reporting summary
Further information on research design is available in the Nature Portfolio Reporting Summary linked to this article.

### Data availability
ABCD data are released publicly on an annual basis via the National Institute of Mental Health Data Archive (NDA, https://data-archive.nimh.nih.gov/abcd). Details about the study can be found at https://abcdstudy.org/. In this application, we used data from the ABCD 4.0 National Data Archive release (NDAR DOI:10.15154/1523041). The simulation results are provided in the Supplementary Data 1 and Supplementary Data 2. The performance of the feature selection is provided in the Supplementary Data 3. The empirical results can be found in the Supplementary Data 4.

### Code availability
The code associated with this research is available on GitHub at https://github.com/junting-ren/ZIV. This repository contains all necessary code and instructions to replicate the analyses described in this paper. The version of ZIV used in this publication was deposited at [61].

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

## Acknowledgements
This work was supported by grant R01MH122688, RF1MH120025, and R01MH128959 funded by the National Institute for Mental Health (NIMH). The ABCD Study is supported by the National Institutes of Health and additional federal partners under award numbers: U01DA041022, U01DA041028, U01DA041048, U01DA041089, U01DA041106, U01DA041117, U01DA041120, U01DA041134, U01DA041148, U01DA041156, U01DA041174, U24DA041123, and U24DA041147. A full list of supporters is available at https://abcdstudy.org/federal-partners.html. A listing of participating sites and a complete listing of the study investigators can be found at https://abcdstudy.org/consortium_members/. ABCD consortium investigators designed and implemented the study and/or provided data but did not necessarily participate in the analysis or writing of this report. This manuscript reflects the views of the authors and may not reflect the opinions or views of the NIH or ABCD consortium investigators.

## Author contributions
Junting Ren: Conceptualization, Methodology, Software, Writing - original draft, Visualization. Robert Loughnan: Writing - Review & Editing. Bohan Xu: Software. Wesley K. Thonpson: Conceptualization, Writing - review & editing, Supervision, Funding acquisition. Chun Chieh Fan: Conceptualization, Writing - review & editing, Visualization, Supervision, Funding acquisition.

## Competing interests
The authors declare no competing interests.
