## [Peer Review File · Communications Biology]

Reviewers' comments:

Reviewer #1 (Remarks to the Author):

The manuscript by Ren and colleagues addresses an important issue in modern brain-imaging analyses, in that it provides a Bayesian method that can analyse “zero inflated outcomes” that are very common for example in questionnaire scores. The manuscript builds on the existing literature, that proposed methods (based on linear mixed models – LMM) to estimate the fraction of variance explained (FVE) by brain measurements. The proposed method (ZIV) is suited to zero-inflated variance, but also estimate the proportion of non-null effect, which is very informative to apprehend the complexity of brain-trait associations and can guide data collection and analyses. This proportion cannot be estimated using the current methods, and can be of significant impact for the field. The manuscript performs extensive simulations to validate the method, and demonstrate its benefit against existing LMM. In addition, the authors apply their method to the ABCD data.

For context, the Bayesian approach that the authors implemented, has been used in the field of genetics (so have LMM), where it has also demonstrated great utility. I found the manuscript well written, very clear, and sound. The results are also consistent with what has been observed in genetics.

My suggestions/comments are mostly about clarifications and (small) additional analyses if the authors judge them pertinent.

Major/Less minor:

- In my opinion, Table 1 would benefit from being a figure, which would make the case for ZIV more compelling. It would also help showing the confidence intervals around estimates. I was not sure how to read the Error rates – or how they related to 95%CI coverage. Overall, a discussion on CI (hence power) of the ZIV approach would be of interest. In addition, making Table 1 a figure would also help showing the CIs around the prediction MAE (i.e. is prediction accuracy significantly different).

- Could you to run at least 1 simulation scenario for non-truncated traits (i.e. non 0-inflated) – to confirm ZIV and LMM both perform similarly? It would be a good way to compare CIs/power.

- For the readers less used to Bayesian modelling, could you add a sentence on how you would handle multiple testing correction in FVE/PNN estimation.

- On the ABCD analysis. The FVE seem similar from the different modalities, so I wonder if they are they all tagging the same information, or if they tag complementary variance? It would be interesting to to fit a model with all modalities at once and get FVE by all brain measurements. Can you explain why the PNN CIs are larger for rsfMRI than for the other modalities ? Could it be due to the distribution of brain rsfMRI brain feature or number of brain feature?

Are CIs on par with what is observed in simulations?

- Figure 2 C, does not show subcortical structures. Were they not included in the analyses? In particular, what about the hippocampus, which has been associated with pretty much every psychiatric domain in analyses from the ENIGMA consortium. Table A1 indicates 1186 structural brain measurements, are you only showing a subset, if yes, how was it selected?

- Can ZIV scale up to a vertex/voxel wise approach? Can you extrapolate the computation/RAM requirement it would require, say with 100,000 brain features?

- Is ZIV prediction yielding normal predicted traits (i.e. predicts liability) or the predictors have a 0-inflated distribution? If ZIV based prediction is non-normal, it would be fair to truncate the BLUP and LASSO, using a proportion learned in training to see if that can explain the differences in MAE. It would be ideal to also show some prediction on the real ABCD scales.

- Could you add references to specific supplemental sections – otherwise it can be hard to link them to the main text.

- In Supplemental A2 – FDR > 5% for most scenarios. Esp. when FVE is small, as in real data from ABCD. Does this mean that the specific ROI associations you present are likely to contain false positive? Can you implement an effective control of FDR (or FWER)? In light of this, and the number of associations tested, can you safely conclude about the associations with Default-Mode-Networks and Dorsal attention?

- For each of your modalities, $p < N$, meaning it is possible to fit a multiple regression, and to report the R^2 , analogous to your FVE. Multiple regression may not behave perfectly with 0-inflated outcomes, but its advantage is that it does not assume a specific distribution of ROI-trait betas. This could help test if the spike and slab distribution of effects assumed in ZIV is fair. In practice, the “true” distribution may be a mixture of normals, or contain a few ROI with large effects, and the ZIV estimates might also be biased.

Minor:

- I found this sentence of the abstract unclear, as it may read that there are no methods to estimate FVE. I am not sure what you mean by “formidable”.

Modeling these challenging distributions is exacerbated by the absence of statistical models capable of characterizing the total signal attributed to whole-brain imaging features, making the accurate assessment of brain-behavior relationships particularly formidable.

- I would suggest to also mention ZIV can be used to build predictor. Otherwise, the following sentence comes a bit out of the blue

In simulations, ZIV outperformed other linear prediction algorithms.

- I found those 2 sentences unclear in the introduction, you may want to elaborate what you mean by “sparse” or “ubiquity of effects”, and which other models you are criticizing here.

For example, it is unclear if the brain features for specific MRI modalities and behavioral outcomes are in fact not sparse

Moreover, the typical assumption of “ubiquity of effects”

I am not familiar with this term – please define, a normal distribution does not force all brain features to have the same effect

- Maybe rephrase as an hypothesis or add a reference – possibly to a genetics paper?

Misspecifying semi-continuous outcomes as being normally distributed, especially in highly zero-inflated and/or right-skewed data such as these, can cause severe, generally downward bias in brain-behavior estimates

- It would be useful to add a bit more details about the ROI measurements included, maybe in the supplementary? For example, which atlas was used to define ROI, and what measurements were considered. Also, how do you relate ROIs to networks in fMRI, I believe there are several definitions of the RS-networks?

We investigated the relationship between the psychopathology and Region of Interest (ROI) brain measurements, one imaging modality at a time

- If I understand well, you used GCTA for your LMM benchmark (and BLUP). I would suggest to make this clearer, as it sometimes suggests you implemented your own LMM/REML algorithm. You could also add a paragraph or appendix on how you did this in practice, as GCTA expects SNP data, but can be tricked into fitting models on other type of data. Did you also use GCTA to fit/train the LASSO and RIDGE?

Reviewer #2 (Remarks to the Author):

Ren et al propose a zero inflated Bayesian model for modeling behavioral outcomes that can be bounded below. My main concerns are the appropriateness of the content for the readership of communications biology. This reads as a statistics paper (i.e. it is too statistics/math heavy for the general reader), is not formatted in the typical format for communications biology (Introduction, results, discussion methods), and does not provide enough context within biology or address a biological question (the main contribution is the statistical method). My recommendation would be to refer this to a biostatistics or applied statistics journal. If it is considered for publication, it needs substantial revision to discuss and motivate for the goal of the analysis to quantify brain-behavior associations and some impactful conclusions of the findings.

1. The introduction could use more context.

a. The statement “For example, it is unclear if the brain features for specific MRI modalities and behavioral outcomes are in fact not sparse.” What is the importance of sparsity? Why are you

mentioning it here? There is one sentence and then it's not discussed further.

b. "Ubiquity of effects" is undefined. What variance component models are you talking about?

Random effects models? How are these used in imaging to quantify variance explained? When is variance explained used? When modeling using machine learning models?

2. In section 2.1, the model is presented, but the context is not framed within imaging. I assume the features are imaging data, but are they voxel level, region level, edge level? What is the scale of the data your model is designed for?

3. Because the proportion of zeros is modeled with a latent variable that also represents the expressed data, it means that the proportion of zeros is related to the distribution of the nonzero values as well. This is less flexible than classical ZI model such as the ZIP or ZINB, which model the zero component using a separate part of the likelihood. Commenting on or addressing this limitation/feature of your model is critical.

4. The notation is a bit confusing.

a. In the distribution for β_j , δ_0 depends on β_j , but it is just a point-mass prior at zero, so that dependence doesn't make sense to me.

b. What is the point of the reparameterization? Why even start with β_j if it will be defined in terms of $\tilde{\beta}_j$?

c. In the variational posterior (q), the formula for variance of $\tilde{\beta}_j$ is just equal to $\sigma^2_{\beta_j}$, right? Looks like there should be a σ^2_{β} there instead... Is that the variance of the "null" betas? Would be useful to describe it that way. Is that just a prior parameter, or do you obtain the posterior for that?

d. The π_{β_j} is undefined up to this point. What role does π play? It seems like it factors with respect to the parameters $\tilde{\beta}_j$ (because their sparsity seems to be controlled by π_{β_j}), so how does it affect the sparsity of the parameter estimation?

e. Are the β_j assumed to be independent? This assumption is unrealistic.

f. Why is this a variational approximation, because the β_j are assumed to be independent? would be helpful to describe.

g. Why even use a variational approach instead of just assuming that q is the true posterior?

5. Is the variance in equation (4) conditional on X ? Equation (4) is just the variance of the following equation, so it make sense to define the equation after (4) first.

6. "Then divided by 2" should be "divided by two." Why do you need to divide it by two?

7. "In order to obtain a point prediction, we directly take the mean of the coefficient posterior distributions and product sum with the corresponding features." Is that equivalent to \tilde{r}_i ? Is there an equation you can refer to in order to help the reader?

8. Need more details on existing methods (libability-based linear mixed effects models) in the introduction and possibly other places in the paper.

9. What were the settings on the prior parameters? It is not detailed in the paper?

10. What was the distribution of the outcome used for the simulations? How many zeros? For the boundary/zero value, do you just shift the outcome so that the minimum value is zero?

11. What if there's zero at the top and bottom of the scale, or just the top? Can your method be applied there?

Minor:

1. “mophemtricity,” typo?
2. “applied to the neuroimaging data” should be “applied to neuroimaging data”
3. widely-used tool utilized
4. Typo in point estimate in table.
5. What was the sample size for the simulations?

In this work, Ren et al. proposed a Bayesian model aiming to provide robust characterization of the relationship between a psychological/behavior outcome and high-dimensional brain features, especially in the existence of excessive zeros in the outcome. The model is rooted in the settings of linear models and cross-sectional data, and offers global signal detection (by proposing two measures “FVE” and “PNN”) as well as localized inferences about feature importance ranking/selection, which ideally are helpful features in large-scale brain-behavior studies. However, as a statistical paper, the necessities for developing this method need further justification – the method development is isolated from the practical background with a lack of sufficient discussion/illustration about the consequences of ignoring the existence of excessive zeros in the outcomes, how other people’s works tried to deal with this issue and their possible drawbacks. Furthermore, several concerns raised by the model performance in the simulation studies need to be addressed to sufficiently convince people about the model performance before the publication of this work.

Major issues:

By reading the introduction, the method development looks isolated from the practical background. The discussion about this kind of semi-continuous/zero-inflated outcome stopped short after talking about its existence in many types of studies. The authors need to further justify the necessity for developing this method for zero-inflated outcomes by (perhaps) discussing more about the consequences of ignoring the zero-inflated outcomes and treating them as regular (or normally distributed) outcomes. Furthermore, there is a lack of sufficient discussion/review of how other studies tried to deal with this issue (e.g., Tobit 1958 *Econometrica*, Moulton 1995 *Biometrics*, and Duan 1983 *JBES*, etc) and the added value or strength of the proposed model in this paper should be discussed.

The simulation results also raised several outstanding concerns, which undermined my trust in the validity of the method’s performance and the following empirical findings in the ABCD study. First, the unbiasedness of the FVE and PNN estimators from the ZIV model is not convincing. In the simulation results (Section 3.1), the authors said “The ZIV model provides unbiased estimates of the FVE’s under all scenarios (Table 1)”. However, the ZIV still tends to slightly underestimate the true FVE according to Table 1 in all these scenarios, although it outperforms the GCTA. The simulation results are not convincing to me in terms of the unbiasedness. It would be more accurate or prudent to conclude with its consistency if the author could show the bias approaches zero as the sample size increases in these scenarios. Moreover, it’s not clear what the sample size is for the simulation results in Table 1 – is it the resampled ABCD of its original size? In addition, the PNNs inferred by ZIV are also severely underestimated (in other words, severely conservative), but the authors even didn’t mention this in section 3.1 except just saying “Furthermore, unlike GCTA, our ZIV model provides estimates for the proportion of non-nulls.”. This also implies the PNNs

estimated in the subsequent section 3.2 are all smaller than the truth, making people more likely to conclude “sparse” or rare brain-behavior signals across the brain. This also happened to be what the authors concluded in their application to ABCD (Discussion, the 3rd to the last paragraph), which makes people question whether it’s a result of the method’s defect or the truth.

Second, it’s problematic to use lower error rates of credible intervals (CIs) as a measure of good CI construction procedure. When constructing 95% CIs, we are looking for exactly a 95% probability that the true values lie in the intervals. In other words, the error rates are expected to be exactly 5%. Having a lower error rate doesn’t always mean better. Conversely, a 95% CI with a higher actual coverage is a wider and more conservative CI than necessary for the nominal credible level. Such conservative CIs may decrease the statistical power to detect a meaningful association in general. Given the spotted concerns about simulation studies and results, the subsequent application in the ABCD studies is also questionable. This proposed method needs to show reliable performance using appropriate/correct performance measures before its application in real datasets.

Minor issues:

Section 2.4: “..., where k is equal to the mean estimate of π multiplied by the total number of features then divided by 2.”. The rule seemed to be arbitrary – why divided by 2? As the authors mentioned the rules should depend on actual situations the researchers have, I don’t see the point of emphasizing this specific rule. You may just want to use it as a suggestion but still need to explain why divide it by two.

Page 5, formula 1-3: why is the equation for ELBO not a function of X and z , as the function on page 4 indicated? Because you said, “For simplicity, we assume that the feature matrix X is fixed and always conditioned”? I don’t think the simplifying is necessary and it simplified a lot. I suggest you add X and z back to the formula as it’s now kind of confusing.

Page 6, section 2.6: “i.e.,” instead of “i.e.”

Table 1: it would be better to make them in the same order: put the ZIV on the first line in each section of the table. The numbers shown in the subsection of “95% CI Coverage” are not coverage rates. It would be more straightforward to just put the coverage rates there instead of error rates to align with this subsection’s title – or you can change the subsection title.

Simulation study designs: what’s the normal distribution of noise? Standardized normal?

Page 8. Explain the reasons for feature standardization before modeling.

Response to Reviewers’ comments on the manuscript “Estimating the Total Variance Explained by Whole-Brain Imaging for Zero-inflated Outcomes”

Junting Ren, Robert Loughnan, Bohan Xu, Wesley K. Thompson and Chun Chieh Fan

Responses to Reviewer #1

Overall opinion

The manuscript by Ren and colleagues addresses an important issue in modern brain-imaging analyses, in that it provides a Bayesian method that can analyse “zero inflated outcomes” that are very common for example in questionnaire scores. The manuscript builds on the existing literature, that proposed methods (based on linear mixed models – LMM) to estimate the fraction of variance explained (FVE) by brain measurements. The proposed method (ZIV) is suited to zero-inflated variance, but also estimate the proportion of non-null effect, which is very informative to apprehend the complexity of brain-trait associations and can guide data collection and analyses. This proportion cannot be estimated using the current methods, and can be of significant impact for the field. The manuscript performs extensive simulations to validate the method, and demonstrate its benefit against existing LMM. In addition, the authors apply their method to the ABCD data. For context, the Bayesian approach that the authors implemented, has been used in the field of genetics (so have LMM), where it has also demonstrated great utility. I found the manuscript well written, very clear, and sound. The results are also consistent with what has been observed in genetics. My suggestions/comments are mostly about clarifications and (small) additional analyses if the authors judge them pertinent.

Response: We are sincerely grateful for your insightful and constructive feedback on our manuscript. Your acknowledgement of the significance and clarity of our work, in developing a novel Bayesian method for analyzing zero-inflated outcomes in brain-imaging analyses, is immensely encouraging. Additionally, we thank you for highlighting areas in our manuscript that would benefit from further clarification. We value your expertise and have carefully addressed each of your comments, as detailed below.

Major Comments

(1) *In my opinion, Table 1 would benefit from being a figure, which would make the case for ZIV more compelling. It would also help showing the confidence intervals around estimates. I was not sure how to read the Error rates – or how they related to 95%CI coverage. Overall, a discussion on CI (hence power) of the ZIV approach would be of interest. In addition, making Table 1 a figure would also help showing the CIs around the prediction MAE (i.e. is prediction accuracy significantly different).*

Response: Thank you for your insightful suggestion regarding the presentation of our results. In response, we have restructured Table 1 into three distinct figures to enhance clarity and visual appeal: Figure 2 now presents the point estimates and prediction error, Figure 3 focuses on the coverage and range of credible intervals (CIs), and Figure 4 addresses feature selection sensitivity and the false discovery rate (FDR). Where applicable, we have incorporated error

bars representing one standard deviation of the multiple simulation instances in these figures to denote if the differences among various models are statistically significant.

Furthermore, we have expanded our manuscript to include a detailed discussion on the implications and significance of credible intervals:

Section 3:

“The CI ranges for FVE decreases as sample size increases for both models, whereas the CI ranges for PNN stays relatively constant. Although the coverage rate is higher than nominal level for some setup, the relatively small range of the CI, comparing to the original scale of the estimates, still offers meaningful information of the point estimates.”

Section 4:

“For simulations utilizing real tfMRI features, the model slightly underestimates FVE and inconsistently estimates the PNN: overestimating at lower true values and underestimating at higher ones. The FVE underestimation is linked to the zero-inflation censoring, leading to incomplete information, while the PNN inconsistency arises from the high correlation among tfMRI features, necessitating wider credible interval to address this uncertainty. These findings underscore the need for further investigation to improve the estimation and inference processes related to PNN in future studies. Despite these challenges, it is important to highlight that the ZIV model, when compared to existing methodologies such as GCTA, offers superior performance. This is evident in the higher accuracy in FVE and prediction. Moreover, the ZIV model offers the advantage of providing PNN estimation, enabling uncertainty quantification with credible intervals, and provide method to select important features with high sensitivity and low FDR.”

We believe these revisions and the added discussions significantly strengthen our manuscript by providing a clearer, more comprehensive understanding of the ZIV approach’s efficacy and its advantages over traditional methods. Thank you again for your constructive feedback.

(2) *Could you to run at least 1 simulation scenario for non-truncated traits (i.e. non 0-inflated) – to confirm ZIV and LMM both perform similarly? It would be a good way to compare Cis/power.*

Response: We appreciate your valuable suggestion regarding the comparison between the ZIV and GCTA. In accordance with your request, we have conducted additional simulations for non-truncated (non-zero-inflated) traits, the results of which are illustrated in Figure 2A. This comparison reveals that the performance of ZIV and LMM in estimating the FVE is comparable, particularly when the sample size exceeds the number of features in outcomes that are not zero-inflated.

(3) *For the readers less used to Bayesian modelling, could you add a sentence on how you would handle multiple testing correction in FVE/PNN estimation.*

Response: Thank you for your valuable suggestion. Recognizing the importance of this issue, we have added a comprehensive paragraph in the Discussion section to elucidate the multiple testing correction inherent in our Bayesian model:

“In addition, our Bayesian model inherently incorporates multiple testing correction by estimating the PPN π in the model George and McCulloch [1993], which indicates that the proportion of true causal features is likely to be less than π . Consider the simulated scenario where the true PNN and FVE is set at 0.05 and 0.8 respectively, as shown in Figure 2B. Our model’s inferred PNN is 0.049, which suggests that the proportion of features with a positive association is below 5%. Consequently, when selecting significant features, we would identify fewer than 5% of them as having a high probability of being non-null, based on each feature’s local posterior non-null probability. Figure 4 demonstrates that this approach achieve high sensitivity and low FDR under various setup. It contrasts with the multiple testing strategy used in Genome-Wide Association Studies (GWAS), where each feature is evaluated independently using univariate regression Uffelmann et al. [2021]. In GWAS, features are deemed significant based on their individual p-values, without an aggregate estimate of the proportion of non-null features across the entire set.”

(4) *On the ABCD analysis. The FVE seem similar from the different modalities, so I wonder if they are they all tagging the same information, or if they tag complementary variance? It would be interesting to to fit a model with all modalities at once and get FVE by all brain measurements. Can you explain why the PNN Cis are larger for rsfMRI than for the other modalities ? Could it be due to the distribution of brain rsfMRI brain feature or number of brain feature? Are Cis on par with what is observed in simulations?*

Response: Indeed, the FVE estimates for a given domain score are remarkably similar across imaging modalities. As we subsequently shown in the Figure 5 B and C, the posteriors were converging on the attention related networks, including temporal-parietal, dorsal-lateral, and cingulates. The converging results across modalities implicate a common biological signals. Nevertheless, ZIV currently only model one latent variable per outcome measures and the estimates become unstable when the sample size are much smaller than the number of independent variables. Therefore, we cannot perform a formal test on the signal overlapping between modalities for a given outcome measure.

In our simulations, we did not observe the change of CI as a function of the number of independent variables. The sample size and the underlying signal architecture are more determining factors for the uncertainties in the estimates. In the empirical applications, we also did not observe the differences in CI as a function of number of features in each imaging modality except resting state fMRI. Although the CI shown in the simulations (Figure 2) and the CI shown in the empirical applications (Figure 5) are not directly comparable, because one was derived some repeated simulations and the other was derived from approximation by Variational Bayes, both seems to be more related to the underlying signal architectures rather than the number of input features. The correlations among input features would have impact on the estimates as well, as we see the differences between the simulations with limited correlations and empirical correlations. It is possible that the correlations among features of resting state fMRI can overshadow the PNN estimates, leading to higher uncertainty. The complex interplay of the measurement noises, feature correlations, and true signal architecture would require a much more complex modeling strategy to test. ZIV is just a very first step toward to the direction.

Accordingly, we added the notes to the observations raised by the reviewer in our result and discuss section. In the ABCD Study results:

“Across CBCL scales, the estimated FVE ranges from 0.6% to 4.9%. Imaging modalities have a similar range of FVE’s given the same CBCL subscale, with the highest magnitude of FVE in the DSM-oriented Conduct subscale and the Rule Break syndrome subscale. The proportion of non-nulls also has similar range across all imaging modalities except rsfMRI. Across CBCL subscales, rsfMRI has the highest proportion of non-null compared to other imaging modalities. It also has the largest model uncertainties in PNN comparing to others.”

In the discuss section:

“When simulating zero-inflated outcomes with independent features, we observe that point estimates progressively align with the ground truth as the sample size increases, supporting the model’s statistical consistency. For simulations utilizing real tfMRI features, the model slightly underestimates FVE and inconsistently estimates the PNN: overestimating at lower true values and underestimating at higher ones. The FVE underestimation is linked to the zero-inflation censoring, leading to incomplete information, while the PNN inconsistency arises from the high correlation among tfMRI features, necessitating wider credible interval to address this uncertainty. These findings underscore the need for further investigation to improve the estimation and inference processes related to PNN in future studies. Because we do not explicitly assign different priors for the input subset and do not include bivariate outcomes, we cannot test the signal overlaps between imaging modalities. It remains to be seen if the converging results we found across modalities in the ABCD data are indeed tagging the same biological signals.”

(5) *Figure 2 C, does not show subcortical structures. Were they not included in the analyses?*

In particular, what about the hippocampus, which has been associated with pretty much every psychiatric domain in analyses from the ENIGMA consortium. Table A1 indicates 1186 structural brain measurements, are you only showing a subset, if yes, how was it selected?

Response: We included all region-of-interest (ROIs) from the ABCD data release, therefore, the measurements on the subcortical structures, such as hippocampus were in the model as well. Nevertheless, none of the functional or structural measurement on the hippocampus has shown evident associations with the CBCL items in the posterior estimates. The posterior effect sizes were concentrated on the cortical structures, therefore, we only render the results on the cortical surface.

As the reviewer sagely pointed out, we were surprised by the lack of associations on hippocampus as well. Nevertheless, our analyses were different from the ENIGMA in several aspects. First, our inference is based on the entire set of ROIs, one modality-at-a-time, instead of an univariate, one ROI-at-a-time, approach. It means that the algorithm assign the probability of being causal based on the signal patterns across whole brain regions, hence reducing the apparent signals from the hippocampus if the nearby structure has stronger and more consistent associations with the outcome measures. We observed increased signals in the parahippocampus and the entorhinal cortex, as shown in the Figure 5 C and 5 D. Second, our analysis is based on the adolescents from a general population cohort that are not enriched for psychiatric disorders. It means our analyses would be more sensitive to the prevalent symptoms among youth, rather than broad spectrum of severe psychiatric disorders.

We provide the full result in the supplementary information and add the following paragraph to the discussion section:

“Surprisingly, subcortical structures, such as hippocampus, do not exhibit stronger signals than other brain regions in our analyses. The posterior effect sizes are concentrated on the cortical regions. While such observations are consistent with prior reports based on youth samples Ducharme et al. [2012], Chabernaud et al. [2012], Whitfield-Gabrieli et al. [2020], it is in sharp contrast to the results from the ENIGMA consortium Opel et al. [2020]. The meta-analyses on the case-control designed studies show most of the consistent differences between patients with psychiatric disorders and healthy controls are located at hippocampus Opel et al. [2020]. It is possible what we capture here, is the neural substrate underlying normal variations of the psychopathologies among youth or the early precursor of a disease process. Our application of ZIV is the very first step toward the understanding of the etiology of psychiatric disorders. ”

(6) *Can ZIV scale up to a vertex/voxel wise approach? Can you extrapolate the computation/RAM requirement it would require, say with 100,000 brain features?*

Response: The number of features has limited impact on the computation/RAM requirement, as the computational complexity of ZIV is scaled by number of the subjects. However, our current implementation has stable estimates when the number of features is not overwhelmingly exceeding the number of subjects. Based on this observation, we do not recommend to use ZIV on vertex/voxel wise analysis.

(7) *Is ZIV prediction yielding normal predicted traits (i.e. predicts liability) or the predictors have a 0-inflated distribution? If ZIV based prediction is non-normal, it would be fair to truncate the BLUP and LASSO, using a proportion learned in training to see if that can explain the differences in MAE. It would be Ideal to also show some prediction on the real ABCD scales.*

Response: Thank you for your query regarding the nature of predictions made by the ZIV model. We would like to clarify that the predictions made by the ZIV model indeed yield traits on the real ABCD scale, by applying a truncation at zero for values falling below this threshold. To ensure a level playing field and facilitate a meaningful comparison across models, we have similarly adjusted the predictions from LASSO, Ridge, and GCTA models by truncating their predicted values to zero whenever these values fall below zero. This approach guarantees that the predictions from linear models accurately reflect the zero-inflated nature of the outcomes.

To elucidate this process and its rationale, we have incorporated the following paragraph in Section 4 of our manuscript:

“For a fair comparison of prediction accuracy with linear predictors such as Lasso, Ridge and GCTA, we truncate their predictions to 0 when the values are below zero. This ensures that the linear models takes into account the zero-inflated nature of the outcomes.”

(8) *Could you add references to specific supplemental sections – otherwise it can be hard to link them to the main text.*

Response: Thank you for the suggestion. We have added specific references to the supplemental sections where applicable.

(9) *In Supplemental A2 – $FDR > 5\%$ for most scenarios. Esp. when FVE is small, as in real data from ABCD. Does this mean that the specific ROI associations you present are likely to contain false positive? Can you implement an effective control of FDR (or FWER)? In light of this, and the number of associations tested, can you safely conclude about the associations with Default-Mode-Networks and Dorsal attention?*

Response: To provide a better guide on the selection of threshold for statistical inference, we conduct additional simulations to examine the sensitivity and false discovery rate (Figure 4). We found that, given the eight thousands subjects with eight hundreds correlated imaging features, we can control the false discovery rate less than 10% if we only select features that has local pi higher than 50%. It means that the results we shown in the Figure 5B to 5D are valid with appropriate control of FDR.

(10) *For each of your modalities, $p < N$, meaning it is possible to fit a multiple regression, and to report the R^2 , analogous to your FVE. Multiple regression may not behave perfectly with 0-inflated outcomes, but its advantage is that it does not assume a specific distribution of ROI-trait betas. This could help test if the spike and slab distribution of effects assumed in ZIV is fair. In practice, the “true” distribution may be a mixture of normals, or contain a few ROI with large effects, and the ZIV estimates might also be biased.*

Response: Because the calculation of variance explained (R^2) requires the assumption of the distribution of the outcome variables, we found it hard to map onto the general framework we put here. In our simulations, we show that regardless of the level of sparsity, the FVE is unbiased in both linear and zero-inflated outcomes. It is slightly underestimating the FVE if the correlations among features are high, regardless the true underlying signal structures. Since the metrics we shown in the empirical applications on ABCD Study are not sensitive to the signal architecture and the outcomes are zero-inflated in the observational level, we did not pursuit further analyses using multiple regression framework. Because our model is based on the Bayesian estimator, what we evaluate is the distribution of the parameter estimates. The credible intervals that range in sparsity less than 10% for all imaging modalities across all domain scores suggest the brain-behavioral associations among youth are indeed sparse.

Minor comments

(a) *I found this sentence of the abstract unclear, as it may read that there are no methods to estimate FVE. I am not sure what you mean by “formidable”. Modeling these challenging distributions is exacerbated by the absence of statistical models capable of characterizing the total signal attributed to whole-brain imaging features, making the accurate assessment of brain-behavior relationships particularly formidable.*

Response: We revised our abstract accordingly:

“There is a dearth of statistical models that adequately capture the total signal attributed to whole-brain imaging features. The total signal is often widely distributed across the brain, with

individual imaging features exhibiting small effect sizes for predicting neurobehavioral phenotypes. The challenge of capturing the total signal is compounded by the distribution of neurobehavioral data, particularly responses to psychological questionnaires, which often feature zero-inflated, highly skewed outcomes. To close this gap, we have developed a novel Variational Bayes algorithm that characterizes the total signal captured by whole-brain imaging features for zero-inflated outcomes. Our zero-inflated variance (ZIV) estimator robustly estimates the fraction of variance explained (FVE) and the proportion of non-null effects (PNN) from large-scale imaging data. In simulations, ZIV demonstrates superior performance over other linear models. When applied to data from the Adolescent Brain Cognitive DevelopmentSM (ABCD) Study, we found that whole-brain imaging features contribute to a larger FVE for externalizing behaviors compared to internalizing behaviors. Moreover, focusing on features contributing to the PNN, ZIV estimator localized key neurocircuitry associated with neurobehavioral traits. ”

(b) I would suggest to also mention ZIV can be used to build predictor. Otherwise, the following sentence comes a bit out of the blue: “In simulations, ZIV outperformed other linear prediction algorithms.”

Response: Thank you for your valuable suggestion to clarify the application of ZIV in constructing a predictor. We have accordingly expanded the relevant subsection to provide a more detailed explanation:

(Section 2.6) “In our Bayesian model, we can obtain a posterior distribution for z_i through sampling from the posterior distribution multiple times:

$$y_i^{(c)} = \beta_0^{(c)} + \mathbf{x}_i^T \boldsymbol{\beta}^{(c)}$$

Then,

$$z_i^{(c)} = \begin{cases} y_i^{(c)} & , y_i^{(c)} > 0 \\ 0 & , y_i^{(c)} \leq 0 \end{cases}$$

where $\beta_0^{(c)}$ and $\boldsymbol{\beta}^{(c)}$ are the sampled parameters from their corresponding posterior distribution at cycle c .

In order to obtain a point prediction, we directly take the mean of the coefficient posterior distributions and product sum with the corresponding features:

$$y_i = \mathbb{E}[\beta_0] + \mathbf{x}_i^T \mathbb{E}[\boldsymbol{\beta}]$$

$$z_i = \begin{cases} y_i & , y_i > 0 \\ 0 & , y_i \leq 0 \end{cases}$$

This is equivalent to sample z_i from the posterior and take the mean of the sampled z_i , due to the linearity of the expectations.

For a fair comparison of prediction accuracy with linear predictors such as Lasso, Ridge and GCTA, we truncate their predictions to 0 when the values are below zero. This ensures that the linear models takes into account the zero-inflated nature of the outcomes.”

(c) I found those 2 sentences unclear in the introduction, you may want to elaborate what you mean by “sparse” or “ubiquity of effects”, and which other models you are criticizing here. For example, it is unclear if the brain features for specific MRI modalities and behavioral outcomes are in fact not sparse. Moreover, the typical assumption of “ubiquity of effects” I am not familiar with this term – please define, a normal distribution does not force all brain feature to have the same effect.

Response: We thank the reviewer’s comment on our use of the jargon in the original submission. Here, we revised the introduction section substantially while replacing the jargon with more accessible plain descriptions.

(d) *Maybe rephrase as an hypothesis or add a reference – possibly to a genetics paper? Misspecifying semi-continuous outcomes as being normally distributed, especially in highly zero-inflated and/or right-skewed data such as these, can cause severe, generally downward bias in brain-behavior estimates*

Response: We revised this sentence and added one relevant reference from the genetic field: *“Misspecifying semi-continuous outcomes as being normally-distributed would lead to incorrect specification on the variance of the outcomes, generally leading to downward bias in the estimation on total variance explained [N. Fusi and Stegle, 2014].”*

(e) *It would be useful to add a bit more details about the ROI measurements included, maybe in the supplementary? For example, which atlas was used to define ROI, and what measurements were considered. Also, how do you relate ROIs to networks in fMRI, I believe there are several definitions of the RS-networks? We investigated the relationship between the psychopathology and Region of Interest (ROI) brain measurements, one imaging modality at a time*

Response: We added those information in the supplementary and rearrange the method section to provide a better clarity on the imaging features.

(f) *If I understand well, you used GCTA for your LMM benchmark (and BLUP). I would suggest to make this clearer, as it sometimes suggests you implemented your own LMM/REML algorithm. You could also add a paragraph or appendix on how you did this in practice, as GCTA expects SNP data, but can be tricked into fitting models on other type of data. Did you also use GCTA to fit/train the LASSO and RIDGE?*

Response: We added paragraphs about the GCTA, and also the way Lasso and Ridge got fit to improve the accessibility in the method section: *“GCTA expects the lower triangular part of the relatedness matrices as the input for estimating FVE. We calculated the relatedness matrix given the correlations across input imaging features [Couvry-Duchesne et al., 2020, Zhang et al., 2019], and then call GCTA to perform the estimation given the imaging-based relatedness matrix. In addition to FVE, GCTA also output BLUP for the prediction. We also implemented Ridge regression and Lasso, using scipy, to compare the prediction accuracy across models, utilizing mean absolute error (MAE) as the metric.”*

Responses to Reviewer #2

Overall opinion

Ren et al propose a zero inflated Bayesian model for modeling behavioral outcomes that can be bounded below. My main concerns are the appropriateness of the content for the readership of communications biology. This reads as a statistics paper (i.e. it is too statistics/math heavy for the general reader), is not formatted in the typical format for communications biology (Introduction, results, discussion methods), and does not provide enough context within biology or address a biological question (the main contribution is the statistical method). My recommendation would be to refer this to a biostatistics or applied statistics journal. If it is considered for publication, it needs substantial revision to discuss and motivate for the goal of the analysis to quantify brain-behavior associations and some impactful conclusions of the findings.

Response: We are grateful for the reviewer's thoughtful feedback, particularly concerning the manuscript's scientific motivation and its alignment with the readership of Communications Biology. Acknowledging the reviewer's concerns, we have undertaken substantial revisions, especially within the introduction section, to provide additional scientific motivation. These enhancements aim to more clearly articulate the relevance of our zero-inflated Bayesian model to the biological sciences, emphasizing its potential to elucidate brain-behavior associations. The full text of the revised introduction is presented to reflect these changes, ensuring our manuscript more aptly meets the expectations and interests of Communications Biology's readership.

Major comments

(1) *The introduction could use more context.*

- *The statement "For example, it is unclear if the brain features for specific MRI modalities and behavioral outcomes are in fact not sparse." What is the importance of sparsity? Why are you mentioning it here? There is one sentence and then it's not discussed further.*
- *"Ubiquity of effects" is undefined. What variance component models are you talking about? Random effects models? How are these used in imaging to quantify variance explained? When is variance explained used? When modeling using machine learning models?*

Response: We revise the introduction to provide more context in a plainer language instead of technical short-hands. Regardless of how the model is specified, the key issue is about the balance between inferring causal regions and characterizing the total signals given whole brain features. Prior models only lean on one or the other. ZIV is trying to achieve both.

(2) *In section 2.1, the model is presented, but the context is not framed within imaging. I assume the features are imaging data, but are they voxel level, region level, edge level? What is the scale of the data your model is designed for?*

Response: We clarify the approach we have in both introduction and method section, emphasizing our unit of analysis is on the ROI level.

(3) *Because the proportion of zeros is modeled with a latent variable that also represents the expressed data, it means that the proportion of zeros is related to the distribution of the nonzero values as well. This is less flexible than classical ZI model such as the ZIP or ZINB, which model the zero component using a separate part of the likelihood. Commenting on or addressing this limitation/feature of your model is critical.*

Response: We appreciate your critical assessment of our model's approach to handling zeros and its comparison with classical Zero-Inflation models like ZIP and ZINB. In response to your suggestion, we have incorporated a new paragraph in the discussion section:

“Our model addresses the zero-inflated nature of the data through modeling a single latent variable, whereas traditional two-part models treat the zero as true values and separately describes the probability of the outcome being positive and the magnitude of positive values Lambert [1992], Yau et al. [2003], Ren et al. [2022]. The two-part models offer flexibility by not presuming data below the detection threshold (zero) as unobserved. However, for brain imaging applications, the ZIV model demonstrates greater suitability. First, the latent model provides enhanced interpretability, estimating FVE with easily comprehensible coefficient effects. In contrast, the two-part model does not directly estimate FVE and requires interpreting two distinct sets of coefficients. Second, when considering behavioral outcomes, it is logical to assume the behavior remains unobserved until the latent variable surpasses a certain threshold.”

(4) The notation is a bit confusing.

- (a) In the distribution for β_j , δ_0 depends on β_j , but it is just a point-mass prior at zero, so that dependence doesn't make sense to me.
- (b) What is the point of the reparameterization? Why even start with β_j if it will be defined in terms of $\tilde{\beta}_j$?
- (c) In the variational posterior (q), the formula for variance of $\tilde{\beta}_j$ is just equal to $\sigma_{\beta_j}^2$, right? Looks like there should be a $\sigma_{\tilde{\beta}}^2$ there instead... Is that the variance of the “null” betas? Would be useful to describe it that way. Is that just a prior parameter, or do you obtain the posterior for that?
- (d) The π_{β_j} is undefined up to this point. What role does π play? It seems like it factors with respect to the parameters $\tilde{\beta}_j$ (because their sparsity seems to be controlled by π_{β_j}), so how does it affect the sparsity of the parameter estimation
- (e) Are the β_j assumed to be independent? This assumption is unrealistic.
- (f) Why is this a variational approximation, because the β_j are assumed to be independent? would be helpful to describe.
- (g) Why even use a variational approach instead of just assuming that q is the true posterior?

Response: Thank you for your valuable feedback on the notation and structure of our method section. We have made revisions to enhance clarity, especially for readers less familiar with statistical methodologies. First, the revised method section now includes subsections for a clearer presentation, such as model overview, prior specification, and posterior inference methods... We begin with an accessible explanation of our posterior inference objective in Section 2.3, improving readability for a broader audience:

“The objective of posterior inference is to obtain the posterior distribution of the model parameters, formulated as:

$$\mathbb{P}(\tilde{\beta}, \delta, \pi, \sigma | \mathbf{X}, \mathbf{z}) \propto \mathbb{P}(\mathbf{z} | \tilde{\beta}, \delta, \pi, \sigma) \times \mathbb{P}(\tilde{\beta}, \delta, \pi, \sigma)$$

where $\mathbb{P}(\mathbf{z} | \tilde{\beta}, \delta, \pi, \sigma)$ represents the data likelihood and $\mathbb{P}(\tilde{\beta}, \delta, \pi, \sigma)$ denote the prior likelihood. The posterior distribution encompasses a variety of random variables, including continuous ones like $\tilde{\beta}$ and discrete ones such as δ . Due to this mix of variable types and high dimensionality of the parameters, deriving a closed-form expression of the distribution for direct sampling is impractical.

Therefore, we provide two different estimation algorithms: Markov Variational Inference (VI) and Chain Monte Carlo (MCMC). MCMC, a widely recognized method, constructs a Markov chain for sampling from the posterior distribution, offering precise estimations Gilks et al. [1995]. In contrast, VI, a more recent technique, approximates the posterior with simpler parametric

distributions Blei et al. [2017]. While VI is more efficient in terms of computation and scalability, it tends to offer approximations that are less precise compared to the direct sampling approach of MCMC Blei et al. [2017]. Offering two estimation algorithms provides users with the flexibility to choose based on their specific requirements, whether prioritizing speed or accuracy.”

Addressing your specific comments:

- (a) We made the following revision in Method Section 2.2 to clarify the effect of δ_0 in the prior distribution:

(Section 2.2)

“We model the effects of the features as a mixture of priors from normally distributed non-nulls and point mass nulls (slab and spike prior):

$$\beta_j \sim N(0, \sigma_\beta)\pi + \delta_0(\beta_j)(1 - \pi)$$

where δ_0 denotes a point mass at zero. This formulation implies the following: when $\beta_j = 0$, then $\delta_0(\beta_j) = 1$, leading the density to be the sum of $1 - \pi$ and π times the normal density value evaluated at zero. On the other hand, When $\beta_j \neq 0$, $\delta_0(\beta_j) = 0$, and the density simplifies to π times the normal density at β_j .”

- (b) Thanks for expressing your concern about the reparametrization of β_j into $\delta_j \tilde{\beta}_j$. The reason is that $\delta_j \tilde{\beta}_j$ is not part of the data-likelihood model, it can be expressed as $\delta_j \tilde{\beta}_j$ only after we specified the spike-and-slab prior. Therefore, it is custom to start out with β , which is in align with other papers that utilizes spike-and-slab prior ?George and McCulloch [1993]. We also explained in detail why this reparametrization is beneficial in terms of interpretation of the model:

(Section 2.2)

“Reparameterization can simplify the model’s structure, making it easier to understand and interpret. By expressing β_j as a product of δ_j and $\tilde{\beta}_j$, this effectively separate the mixture model into its components. This decomposition not only facilitates a more intuitive understanding of the model but also simplifies the derivation process in subsequent steps.”

- (c) Variational Posterior distribution (q) has completely different parameters when compared with the prior parameters. Variational Posterior distribution is a family of parametric distribution deliberately simplified to approximate the true posterior distribution so that the computation would be feasible. We have added multiple pieces of information in the manuscript to explain what the variational Posterior distribution (q) is for:

(Section 2.3)

“The objective of posterior inference is to obtain the posterior distribution of the model parameters, formulated as:

$$\mathbb{P}(\tilde{\beta}, \delta, \pi, \sigma \mid \mathbf{X}, \mathbf{z}) \propto \mathbb{P}(\mathbf{z} \mid \tilde{\beta}, \delta, \pi, \sigma) \times \mathbb{P}(\tilde{\beta}, \delta, \pi, \sigma) \quad (1)$$

where $\mathbb{P}(\mathbf{z} \mid \tilde{\beta}, \delta, \pi, \sigma)$ represents the data likelihood and $\mathbb{P}(\tilde{\beta}, \delta, \pi, \sigma)$ denote the prior likelihood.

The posterior distribution encompasses a variety of random variables, including continuous ones like $\tilde{\beta}$ and discrete ones such as δ . Due to this mix of variable types and high dimensionality of the parameters, deriving a closed-form expression of the distribution for direct sampling is impractical...”

(Section 2.3.1)

“We want to approximate the true posterior as shown in Equation (1). To achieve this, we

use the following variational distributions for approximation:

$$\begin{aligned}
q_\phi(\tilde{\beta}, \delta, \pi, \sigma) &= \prod_{j=1}^p N\left(\tilde{\beta}_j \mid \mu_{\beta_j} \delta_j, \sigma_{\beta_j}^2 \delta_j + \sigma_{\beta_j}^2 (1 - \delta_j)\right) \\
&\times \prod_{j=1}^p \text{Bernoulli}(\delta_j \mid \pi_{\beta_j}) \\
&\times \text{Beta}(\pi \mid a_3, a_4) \\
&\times \text{LogNormal}(\sigma^2 \mid \mu = b_3, \sigma^2 = b_4)
\end{aligned}$$

The deliberate decision to maintain independence among these variational distributions is to simplify the estimation algorithm. It's important to note, however, that in reality, the true posterior for the parameters should be correlated: the true posterior of $\tilde{\beta}, \delta, \pi, \sigma$ is a multivariate distribution, which is intractable. Our approach with the variational distribution is to approximate this true posterior using simpler parametric forms. We introduce variational parameters as $\phi = (a_3, a_4, b_3, b_4, \mu_\beta, \sigma_\beta, \pi_\beta)$, where $\mu_\beta = (\mu_{\beta_1}, \mu_{\beta_2}, \dots, \mu_{\beta_p})$, $\sigma_\beta = (\sigma_{\beta_1}, \sigma_{\beta_2}, \dots, \sigma_{\beta_p})$, $\pi_\beta = (\pi_{\beta_1}, \pi_{\beta_2}, \dots, \pi_{\beta_p})$. Once these variational parameters ϕ are learned from data, we can sample from q_ϕ to approximate the true posterior distribution. For instance, to obtain the posterior distribution of δ_j , we simply sample from $\text{Bernoulli}(\pi_{\beta_j})$.

- (d) π is the prior for all δ_j that indicates whether β_j is non-null, whereas π_{β_j} is the estimated posterior parameter for δ_j so that we can sample δ_j from $\text{Bernoulli}(\pi_{\beta_j})$ which serves as the posterior for δ_j by accounting for data. In addition, to clarify the role of δ_j , we rewritten the variational distribution as:

$$\begin{aligned}
q_\phi(\tilde{\beta}, \delta, \pi, \sigma) &= \prod_{j=1}^p N\left(\tilde{\beta}_j \mid \mu_{\beta_j} \delta_j, \sigma_{\beta_j}^2 \delta_j + \sigma_{\beta_j}^2 (1 - \delta_j)\right) \\
&\times \prod_{j=1}^p \text{Bernoulli}(\delta_j \mid \pi_{\beta_j}) \\
&\times \text{Beta}(\pi \mid a_3, a_4) \\
&\times \text{LogNormal}(\sigma^2 \mid \mu = b_3, \sigma^2 = b_4)
\end{aligned}$$

- (e) The priors for β_j are assumed to be independent. However, the true posterior for β_j are not independent, not only can they be correlated with other $\beta_k, k \neq j$, but also other parameters such as δ_k . Our variational distribution q uses independent parametric normal distribution for β_j to approximate the true posterior, whereas our new MCMC algorithm directly sample from the true posterior. For more details, please refer to the Method Section 4.
- (f) Thanks for the suggestions. Therefore, we have added additional motivation for using variational approximation in Section 4.3, as shown in the beginning of our response.
- (g) Yes, you are correct that we can directly sample from the true posterior, instead of using q to approximate the true posterior. The drawback is that MCMC (direct sampling) requires computational resource that is exponential in number of features p , whereas when using VI, it is only linear in terms of p . We have added additional comparison between VI and MCMC in the revised manuscript in Method Section 4 and Result Section 2.

(5) Is the variance in equation (4) conditional on X ? Equation (4) is just the variance of the following equation, so it make sense to define the equation after (4) first.

Response: Thank you for the insightful suggestion on the notation calculating Fraction of Variance Explained (FVE). We have not only clarified the notation for FVE but also for proportion of non-null (PNN):

(Section 2.4)

“We use FVE and PNN as the two key metrics to characterize the global signal architecture of brain-behavior associations.

To estimate the FVE on the latent scale, we use the sampled values of latent linear effects in the posterior draw of the parameters. By definition, the variance captured by the latent linear outcome at sample cycle c is given by

$$\text{var} \left[\tilde{r}_i^{(c)} \mid \mathbf{x}_i \right], \quad (0.1)$$

where $\tilde{r}_i^{(c)}$ is the latent linear outcome value for individual i at the current cycle c

$$\tilde{r}_i^{(c)} = \sum_{j=1}^p x_{ij} \tilde{\beta}_j^{(c)} \delta_j^{(c)}.$$

Then, the estimated variance in the current cycle c is

$$\hat{\text{var}} \left[\tilde{r}_i^{(c)} \mid \mathbf{x}_i \right] = \frac{\sum_i (\tilde{r}_i^{(c)})^2}{N} - \left(\frac{\sum_i \tilde{r}_i^{(c)}}{N} \right)^2.$$

The estimated FVE on the latent scale is at cycle c is

$$FVE^{(c)} = \frac{\hat{\text{var}} \left(\tilde{r}_i^{(c)} \right)}{\hat{\text{var}} \left(\tilde{r}_i^{(c)} \right) + (\sigma^{(c)})^2}$$

where $(\sigma^{(c)})^2$ is the estimate for the noise variance σ^2 at cycle c . We can obtain the posterior distribution for FVE as we repeat the sampling cycle from the posterior distribution for β_j , δ_j and σ^2 .

Inferences for PNN can be accomplished directly using the approximated posterior distribution:

$$\pi \sim \text{Beta}(\pi \mid a_3, a_4).$$

This Bayesian model inherently incorporates multiple testing correction by estimating the PPN π in the model, which indicates that the proportion of true causal features is likely to be less than π . For a more comprehensive discussion and details on this aspect, refer to the Discussion Section.”

(6) “Then divided by 2” should be “divided by two.” Why do you need to divide it by two?

Response: Thanks for the suggestion in clarifying the criteria for feature selection, we have made the following revision:

(Section 2.5)

“For localized feature selection, the model produces posterior distributions $\delta_j \sim \text{Bernoulli}(\pi_{\beta_j})$ that indicates the probability of non-null for each feature j . Customized rules based on π_{β_j} can be utilized to identify likely non-null features. On the global level, the estimation of PPN π suggests that the total number of non-null features should be less than $p \times \pi$, which in turn penalizes the sum of all δ_j should not exceed $p \times \pi$. Therefore, in the Monte Carlo experiments below, we pick the top k highest π_{β_j} as non-null associations, where $k = t * \hat{\pi} * p$ and $\hat{\pi}$ is the mean of PNN π posterior distribution. We set $t = (1, 0.75, 0.5, 0.25)$ to showcase the trade off between false discovery rate and sensitivity (Supplemental Materials). ”

(7) “In order to obtain a point prediction, we directly take the mean of the coefficient posterior distributions and product sum with the corresponding features.” Is that equivalent to \tilde{r}_i ? Is there an equation you can refer to in order to help the reader?

Response: Thank you for the suggestion in terms of readability of the explaining how the model does prediction. We have made the following revision to enhance the clarity of the manuscript:

(Section 2.6)

“For our Bayesian model, we can obtain a posterior distribution for z_i through sampling from the posterior distribution multiple times:

$$y_i^{(c)} = \beta_0^{(c)} + \mathbf{x}_i^T \boldsymbol{\beta}^{(c)}$$

Then,

$$z_i^{(c)} = \begin{cases} y_i^{(c)} & , y_i^{(c)} > 0 \\ 0 & , y_i^{(c)} \leq 0 \end{cases}$$

where $\beta_0^{(c)}$ and $\boldsymbol{\beta}^{(c)}$ are the sampled parameters from their corresponding posterior distribution at cycle c .

In order to obtain a point prediction, we directly take the mean of the coefficient posterior distributions and product sum with the corresponding features:

$$y_i = \mathbb{E}[\beta_0] + \mathbf{x}_i^T \mathbb{E}[\boldsymbol{\beta}]$$

$$z_i = \begin{cases} y_i & , y_i > 0 \\ 0 & , y_i \leq 0 \end{cases}$$

This is equivalent to sample z_i from the posterior and take the mean of the sampled z_i , due to the linearity of the expectations.”

(8) Need more details on existing methods (liability-based linear mixed effects models) in the introduction and possibly other places in the paper.

Response: We sincerely appreciate your valuable suggestion to enrich the introduction with more comprehensive details on liability-based models. In alignment with your feedback, we have introduced an additional paragraph in the start of Method section to address this need:

“The core of ZIV estimator is a Tobit model with spike-and-slab priors imposed on its coefficients. This model bears resemblance to traditional linear regression but is specifically tailored for situations where the dependent variable is zero-inflated by being censored at a certain threshold. ZIV enables the application of standard linear regression methods on uncensored data, while treating observations as censored values, not exact values Amemiya [1984]. The Tobit model is widely used in health outcome research. For instance, it is utilized for self-reported psychometric scales Austin et al. [2000], for exploring the relationship between Everyday Cognition scales and structural neuroimaging Farias et al. [2013], and for examining the link between memory functions and the 5-HT type 4 receptor Haahr et al. [2013]. The spike-and-slab prior is a Bayesian approach to variable selection and coefficient estimation, utilizing a mixture of two distributions: a “spike” representing a point mass at zero for irrelevant features and a “slab” representing a continuous distribution for relevant features. Imposing a spike-and-slab prior on coefficients of a linear model aids in variable selection and mitigates overfitting in high-dimensional settings Ishwaran and Rao [2005]. It has been frequently used in identifying causal variables with various degree of background correlations, such as fine mapping in genetic applications S. Zhao and He [2024]. The spike-and-slab method has also been utilized to improve the interpretation and detection of neural activity in various studies, such as those involving calcium imaging Murphy et al. [2018], electromagnetic brain mapping Nathoo et al. [2014], and functional MRI Yu et al. [2018], Zeng et al. [2022]. Given the goal to identify key neural pathways underlying complex neurobehaviors, e.g. psychopathology among youth, we adopt this strategy in our ZIV implementation. ”

(9) *What were the settings on the prior parameters? It is not detailed in the paper?*

Response: Thank you for expressing your confusion about the prior parameters. Although the prior information was included in the original manuscript, we acknowledge that the information was not as clearly presented as it should have been. To rectify this, we have now dedicated a specific section in the revised manuscript to thoroughly describe the prior settings. This section includes:

(Section 2.2 Prior Specification)

“We model the effects of the features as a mixture of priors from normally distributed non-nulls and point mass nulls (slab and spike prior):

$$\beta_j \sim N(0, \sigma_\beta)\pi + \delta_0(\beta_j)(1 - \pi)$$

where δ_0 denotes a point mass at zero. This formulation implies the following: when $\beta_j = 0$, then $\delta_0(\beta_j) = 1$, leading the density to be the sum of $1 - \pi$ and π times the normal density value evaluated at zero. On the other hand, When $\beta_j \neq 0$, $\delta_0(\beta_j) = 0$, and the density simplifies to π times the normal density at β_j . We reparameterize β_j , $j = 1, \dots, p$, as

$$\beta_j = \delta_j \tilde{\beta}_j$$

where

$$\begin{aligned}\tilde{\beta}_j &\sim N(0, \sigma_\beta^2) \\ \delta_j &\sim \pi^{\delta_j} (1 - \pi)^{1 - \delta_j}\end{aligned}$$

Reparameterization can simplify the model’s structure, making it easier to understand and interpret. By expressing β_j as a product of δ_j and $\tilde{\beta}_j$, this effectively separate the mixture model into its components. This decomposition not only facilitates a more intuitive understanding of the model but also simplifies the derivation process in subsequent steps.

The specification of priors for other parameters in our model is contingent upon the chosen posterior inference methods:

- *For Variational Inference method, we assume the following non-informative prior for the global proportion of non-nulls, π , and the variance of the error terms, σ^2 , as:*

$$\begin{aligned}\pi &\sim \text{Uniform}(0, 1) \\ \sigma^2 &\sim \text{Uniform}(0, +\infty)\end{aligned}$$

We denote by $\theta = (\sigma_\beta, \beta_0)$ the parameters optimized using gradient descent without a variational posterior.

- *For Markov Chain Monte Carlo Inference Method, we assume the following non-informative priors:*

$$\begin{aligned}\pi &\sim \text{Beta}(0.5, 0.5) \\ \sigma^2 &\sim \text{InverseGamma}(0.1, 0.1) \\ \sigma_\beta^2 &\sim \text{InverseGamma}(1, 1) \\ \beta_0 &\sim \text{Normal}(0, \sigma_\beta^2)\end{aligned}$$

(10) *What was the distribution of the outcome used for the simulations? How many zeros? For the boundary/zero value, do you just shift the outcome so that the minimum value is zero?*

Response: Thanks for pointing out the lack of detail regarding the distribution of the simulate outcome. We have made the following amendment in the manuscript:

(Section 2.7)

The non-null effects were generated from a normal distribution with mean 0 and standard deviation of 0.1. The latent outcome was set equal to the linear combination of the features multiplied by the corresponding coefficients adding normally-distributed noise. The observed outcome was truncated at 0 whenever the latent outcome was negative. Because that the mean of the latent outcome is 0, there are around 50% zeros in the outcomes for each simulated instances.

(11) *What if there's zero at the top and bottom of the scale, or just the top? Can your method be applied there?*

Response: Yes, the direction coding is user-defined in the context of the modeling. Therefore, for the scale that has just topped zeros, ZIV can handle it without trouble. However, ZIV does not handle the outcomes with truncations on both sides (both top and bottom of the scale). Empirically, we did not observe many of such coexistence of the zero-inflation and the ceiling effect on the questionnaire outputs in the ABCD Study. Nonetheless, it can be a natural extension for the latent normal model that the ZIV depends on.

Minor comments

(a) *“mophemtricity,” typo?*

Response: Thank you for pointing out this typo. We have corrected this, it should have been: “morphometricity”.

(b) *“applied to the neuroimaging data” should be “applied to neuroimaging data”*

Response: Thank you for highlighting this grammatical detail. We have revised the phrase to “applied to neuroimaging data” as suggested. Additionally, we have conducted a thorough review of the manuscript to ensure the accuracy and consistency of similar expressions throughout the text.

(c) *widely-used tool utilized*

Response: Thank you for highlighting this grammatical detail. We have revised this to be: “The Child Behavior Checklist (CBCL) is a tool widely utilized for evaluating an extensive range of emotional and behavioral problems in children.”

(d) *Typo in point estimate in table.*

Response: Thank you for pointing out this typo. We have corrected the typo in the table for “Point Estimates”.

(e) *What was the sample size for the simulations?*

Response: We appreciate your query regarding the sample size utilized in our simulations. To ensure clarity and ease of reference, we have highlighted the sample size and number of features used in our simulations both within the caption of Figure 2 and in Section 2 of our manuscript as follows:

“For the zero-inflated outcomes generated by using ABCD study tfMRI image features, we utilize the real tfMRI images feature matrix encompassing 8893 subjects and 885 features, and vary the the true FVE and PNN to evaluate model performance across different real-data scenarios characterized by highly correlated features.”

Responses to Reviewer #3

Overall opinion

In this work, Ren et al. proposed a Bayesian model aiming to provide robust characterization of the relationship between a psychological/behavior outcome and high-dimensional brain features, especially in the existence of excessive zeros in the outcome. The model is rooted in the settings of linear models and cross-sectional data, and offers global signal detection (by proposing two measures “FVE” and “PNN”) as well as localized inferences about feature importance ranking/selection, which ideally are helpful features in large-scale brain-behavior studies. However, as a statistical paper, the necessities for developing this method need further justification – the method development is isolated from the practical background with a lack of sufficient discussion/illustration about the consequences of ignoring the existence of excessive zeros in the outcomes, how other people’s works tried to deal with this issue and their possible drawbacks. Furthermore, several concerns raised by the model performance in the simulation studies need to be addressed to sufficiently convince people about the model performance before the publication of this work.

Response: We sincerely thank the reviewer’s comments and suggestions. We performed a major revision on our manuscript to provide more context about our method development. We also performed extensive simulations for benchmarking our algorithm. Details are listed below.

Major comments

(1) *By reading the introduction, the method development looks isolated from the practical background. The discussion about this kind of semi-continuous/zero-inflated outcome stopped short after talking about its existence in many types of studies. The authors need to further justify the necessity for developing this method for zero-inflated outcomes by (perhaps) discussing more about the consequences of ignoring the zero-inflated outcomes and treating them as regular (or normally distributed) outcomes. Furthermore, there is a lack of sufficient discussion/review of how other studies tried to deal with this issue (e.g., Tobit 1958 *Econometrica*, Moulton 1995 *Biometrics*, and Duan 1983 *JBES*, etc) and the added value or strength of the proposed model in this paper should be discussed.*

Response: We acknowledge the reviewer’s insightful critique regarding the initial presentation of our method and its contextual grounding. In response, we have thoroughly revised the introduction to more deeply root our approach within its practical context and to underscore the necessity of addressing zero-inflated outcomes. The full text of the revised introduction is presented to reflect these changes, ensuring our manuscript more aptly meets the expectations and interests of *Communications Biology*’s readership.

In addition, we discussed how other study deal with this issue in the start of the Method section:

“The core of ZIV estimator is a Tobit model with spike-and-slab priors imposed on its coefficients. This model bears resemblance to traditional linear regression but is specifically tailored for situations where the dependent variable is zero-inflated by being censored at a certain threshold. ZIV enables the application of standard linear regression methods on uncensored data, while treating observations as censored values, not exact values Amemiya [1984]. The Tobit model is widely used in health outcome research. For instance, it is utilized for self-reported psychometric scales Austin et al. [2000], for exploring the relationship between Everyday Cognition scales and structural neuroimaging Farias et al. [2013], and for examining the link between memory functions and the 5-HT type 4 receptor Haahr et al. [2013]. The spike-and-slab prior is a Bayesian approach to variable selection and coefficient estimation, utilizing a mixture of two distributions: a “spike” representing a point mass at zero for irrelevant features and a “slab” representing a continuous distribution for relevant features. Imposing a spike-and-slab prior on coefficients of a linear model aids in variable selection and mitigates overfitting in high-dimensional settings Ishwaran and

Rao [2005]. It has been frequently used in identifying causal variables with various degree of background correlations, such as fine mapping in genetic applications S. Zhao and He [2024]. The spike-and-slab method has also been utilized to improve the interpretation and detection of neural activity in various studies, such as those involving calcium imaging Murphy et al. [2018], electromagnetic brain mapping Nathoo et al. [2014], and functional MRI Yu et al. [2018], Zeng et al. [2022]. Given the goal to identify key neural pathways underlying complex neurobehaviors, e.g. psychopathology among youth, we adopt this strategy in our ZIV implementation. ”

(2) The simulation results also raised several outstanding concerns, which undermined my trust in the validity of the method’s performance and the following empirical findings in the ABCD study. First, the unbiasedness of the FVE and PNN estimators from the ZIV model is not convincing. In the simulation results (Section 3.1), the authors said “The ZIV model provides unbiased estimates of the FVE’s under all scenarios (Table 1)”. However, the ZIV still tends to slightly underestimate the true FVE according to Table 1 in all these scenarios, although it outperforms the GCTA. The simulation results are not convincing to me in terms of the unbiasedness. It would be more accurate or prudent to conclude with its consistency if the author could show the bias approaches zero as the sample size increases in these scenarios. Moreover, it’s not clear what the sample size is for the simulation results in Table 1 – is it the resampled ABCD of its original size? In addition, the PNNs inferred by ZIV are also severely underestimated (in other words, severely conservative), but the authors even didn’t mention this in section 3.1 except just saying “Furthermore, unlike GCTA, our ZIV model provides estimates for the proportion of non-nulls.”. This also implies the PNNs estimated in the subsequent section 3.2 are all smaller than the truth, making people more likely to conclude “sparse” or rare brain-behavior signals across the brain. This also happened to be what the authors concluded in their application to ABCD (Discussion, the 3rd to the last paragraph), which makes people question whether it’s a result of the method’s defect or the truth.

Response: We greatly appreciate your detailed comments and concerns regarding the simulation results and their implications for the validity of our method’s performance, as well as the empirical findings in the ABCD study.

First, to address the concern about the consistency of our estimators, we have conducted additional simulations with independent Gaussian features, varying the sample size to illustrate consistency. This approach was chosen over using ABCD features directly to avoid the high correlation issues among selected causal features. The evidence supporting the consistency of our estimator is presented in Figure 2A: “ For FVE estimation using simulated independent standard normal features, as illustrated in the top row of Figure 2A, the FVE estimates from both ZIV and ZIVM models converge to the true value of 0.50 as the sample size increases, for both linear and zero-inflated outcomes. GCTA showed a similar trend in linear outcome, but significantly underestimates the FVE irrespective of whether it is trained on all observations or solely on positive ones. Both the ZIV and ZIVM estimation of PNN gradually converge to the true value as sample size increases as shown in the second row, whereas the GCTA does not provide estimation for PNN.”

Second, regarding the unbiasedness of FVE and PNN estimates, we acknowledge the slight underestimation of FVE and the PNN estimation challenges in the simulations from ABCD data. This has been addressed in the revised results section:

“Across varying true FVE and PNN values, ZIV marginally underestimates FVE by approximately 0.02, while GCTA drastically underestimates it by 50% of the true FVE value. ZIV’s PNN estimates show a tendency to overestimate at lower true PNN values and underestimate at higher ones”

Additionally, we have expanded upon these points in Section 3 of the discussion: “The FVE underestimation is linked to the zero-inflation censoring, leading to incomplete information, while the PNN inconsistency arises from the high correlation among tfMRI features, necessitating

wider credible interval to address this uncertainty. These findings underscore the need for further investigation to improve the estimation and inference processes related to PNN in future studies.”

Third, we have ensured that the sample size and the number of features are clearly stated in Figure 2A and within the results section for greater clarity and transparency.

(3) *Second, it’s problematic to use lower error rates of credible intervals (CIs) as a measure of good CI construction procedure. When constructing 95% CIs, we are looking for exactly a 95% probability that the true values lie in the intervals. In other words, the error rates are expected to be exactly 5%. Having a lower error rate doesn’t always mean better. Conversely, a 95% CI with a higher actual coverage is a wider and more conservative CI than necessary for the nominal credible level. Such conservative CIs may decrease the statistical power to detect a meaningful association in general. Given the spotted concerns about simulation studies and results, the subsequent application in the ABCD studies is also questionable. This proposed method needs to show reliable performance using appropriate/correct performance measures before its application in real datasets.*

Response: We acknowledge your concerns regarding the interpretation of CI coverage rates. To address this, we have expanded our discussion on the over-coverage of CIs in both the results and discussion sections. Specifically, we note that the CI coverage for FVE and PNN reaches the nominal level as sample size increases in new simulations utilizing independent Gaussian features. We recognize that in the ABCD data simulation, the CI coverage for FVE and PNN exceeds the nominal level. However, the relatively small range of these CIs still provides valuable information as shown in Figure 3, in contrast to GCTA, which does not offer uncertainty quantification for FVE.

Section 3

“For uncertainty quantification, both ZIV and ZIVM provide credible intervals (CI) for FVE and PNN estimates. In contrast, GCTA lacks the capability to quantify uncertainty for FVE. As illustrated in the top figure of Figure 3A, the coverage rate of CIs for FVE provided by ZIV and ZIVM aligns with the anticipated 95% nominal level for linear outcomes. However, for zero-inflated outcomes, while ZIVM’s coverage rate converges to the 95% nominal threshold, ZIV’s coverage rate surpasses it. Regarding the PNN coverage rate, both ZIV and ZIVM models exceed the nominal level when the sample size (n) exceeds the number of predictors (p). The bottom figure of Figure 3 displays the range of the CIs. The CI ranges of ZIV are consistently larger than that of ZIVM when $n > p$. The CI ranges for FVE decreases as sample size increases for both models, whereas the CI ranges for PNN stays relatively constant. Although the coverage rate is higher than nominal level for some setup, the relatively small range of the CI, comparing to the original scale of the estimates, still offers meaningful information of the point estimates.”

Section 4

“These findings underscore the need for further investigation to improve the estimation and inference processes related to PNN in future studies. Despite these challenges, it is important to highlight that the ZIV model, when compared to existing methodologies such as GCTA, offers superior performance. This is evident in the higher accuracy in FVE and prediction. Moreover, the ZIV model offers the advantage of providing PNN estimation, enabling uncertainty quantification with credible intervals, and provide method to select important features with high sensitivity and low FDR. ”

Minor comments

(a) *Section 2.4: “. . . , where k is equal to the mean estimate of π multiplied by the total number of features then divided by 2.”. The rule seemed to be arbitrary – why divided by 2? As the authors mentioned the rules should depend on actual situations the researchers have, I don’t see the point of emphasizing this specific rule. You may just want to use it as a suggestion but still*

need to explain why divide it by two.

Response: We thank you for highlighting the ambiguity regarding the feature selection strategy described in Section 2.4. To address this and provide a clearer understanding, we have revised the section to better articulate the feature selection process, and we have introduced additional figures that illustrate the trade-off between sensitivity and FDR as the number of selected features is varied.

“The ZIV model estimates the proportion of all the features being non-null, denoted as PNN (π), and for each individual feature j , estimates the non-null probability, denoted as FNNP (π_{β_j}). At the global level, the estimation of PPN π implies that the total number of non-null features should not exceed $p \times \pi$ where p is the total number of features. Therefore, in the Monte Carlo experiments below, we identify the top k features with highest FNNP π_{β_j} as non-null, where k is defined as $t \times \hat{\pi} \times p$. We set $t = (100\%, 50\%, 25\%)$ to demonstrate the trade off between sensitivity (out of all true non-null features, the percentage selected) and false discovery rate (FDR, out of all selected features, the percentage being true non-null). In Figure 4A, which illustrates outcomes derived from simulations using independent features, no feature was falsely identified as non-null when the percentage t is below than 50%. Additionally, the sensitivity maintains around 50% across various simulation setup. This indicates that the features captured by the model all are true non-null and out of all true non-null features, 50% of them are selected by our model. In Figure 4A for outcomes generated using ABCD tfMRI image features, similar trend can be observed albeit with a higher FDR and lower sensitivity compared to the independent feature simulations. Notably, in the simulated scenario that closely mirrors real data characteristics (true FVE=0.25 and PNN=0.005), setting t to 25% results in an FDR near zero while still capturing about 50% of the true non-null features. This analysis underscores the model’s efficacy in distinguishing between null and non-null features under varying simulation conditions, illustrating its potential applicability in real-world data analysis.”

(b) Page 5, formula 1-3: *why is the equation for ELBO not a function of X and z , as the function on page 4 indicated? Because you said, “For simplicity, we assume that the feature matrix X is fixed and always conditioned”? I don’t think the simplifying is necessary and it simplified a lot. I suggest you add X and z back to the formula as it’s now kind of confusing.*

Response: We have added X and z back in formulas.

(c) Page 6, section 2.6: *“i.e.,” instead of “i.e.”*

Response: Thank you for catching this. We have corrected this in the manuscript.

(d) Table 1: *it would be better to make them in the same order: put the ZIV on the first line in each section of the table. The numbers shown in the subsection of “95% CI Coverage” are not coverage rates. It would be more straightforward to just put the coverage rates there instead of error rates to align with this subsection’s title – or you can change the subsection title.*

Response: Thank you for your suggestion on the presentation of the simulation result. We have restructured Table 1 into three distinct figures to enhance clarity and visual appeal: Figure 2 now presents the point estimates and prediction error, Figure 3 focuses on the coverage and range of CIs, and Figure 4 addresses feature selection sensitivity and FDR. Where applicable, we have incorporated error bars representing one standard deviation of the multiple simulation instances in these figures to denote if the differences among various models are statistically significant.

(e) *Simulation study designs: what’s the normal distribution of noise? Standardized normal?*

Response: This was explained the Method section 4:

The variance of the random Gaussian noise was determined by the empirical variance of the latent mean (linear combination of the features multiplied by the corresponding coefficients) and the pre-defined FVE parameter for simulation.

(f) *Page 8. Explain the reasons for feature standardization before modeling.*

Response: Thank you for mentioning this. We have added the reason in the revised manuscript:
“ *All features were standardized to have zero mean and unit standard deviation before entering them into the ZIV models to improve numerical stability and enhance convergence speed for gradient descent optimization.* ”

References

- T. Amemiya. Tobit models: A survey. *Journal of econometrics*, 24(1-2):3–61, 1984.
- P. C. Austin, M. Escobar, and J. A. Kopec. The use of the tobit model for analyzing measures of health status. *Quality of Life Research*, 9:901–910, 2000.
- D. M. Blei, A. Kucukelbir, and J. D. McAuliffe. Variational inference: A review for statisticians. *Journal of the American statistical Association*, 112(518):859–877, 2017.
- C. Chabernaud, M. Mennes, C. Kelly, K. Nooner, A. Di Martino, F. X. Castellanos, and M. P. Milham. Dimensional brain-behavior relationships in children with attention-deficit/hyperactivity disorder. *Biological Psychiatry*, 71(5):434–442, 2012. ISSN 0006-3223. doi: 10.1016/j.biopsych.2011.08.013. URL <https://dx.doi.org/10.1016/j.biopsych.2011.08.013>.
- B. Couvy-Duchesne, L. T. Strike, F. Zhang, Y. Holtz, Z. Zheng, K. E. Kemper, L. Yengo, O. Colliot, M. J. Wright, N. R. Wray, et al. A unified framework for association and prediction from vertex-wise grey-matter structure. *Human Brain Mapping*, 41(14):4062–4076, 2020.
- S. Ducharme, J. J. Hudziak, K. N. Botteron, M. D. Albaugh, T.-V. Nguyen, S. Karama, and A. C. Evans. Decreased regional cortical thickness and thinning rate are associated with inattention symptoms in healthy children. *Journal of the American Academy of Child Adolescent Psychiatry*, 51(1):18–27.e2, 2012. ISSN 0890-8567. doi: 10.1016/j.jaac.2011.09.022. URL <https://dx.doi.org/10.1016/j.jaac.2011.09.022>.
- S. T. Farias, L. Q. Park, D. J. Harvey, C. Simon, B. R. Reed, O. Carmichael, and D. Mungas. Everyday cognition in older adults: associations with neuropsychological performance and structural brain imaging. *Journal of the International Neuropsychological Society*, 19(4):430–441, 2013.
- E. I. George and R. E. McCulloch. Variable selection via gibbs sampling. *Journal of the American Statistical association*, pages 881–889, 1993.
- W. R. Gilks, S. Richardson, and D. Spiegelhalter. *Markov chain Monte Carlo in practice*. CRC press, 1995.
- M. E. Haahr, P. Fisher, K. Holst, K. Madsen, C. G. Jensen, L. Marner, S. Lehel, W. Baaré, G. Knudsen, and S. Hasselbalch. The 5-HT₄ receptor levels in hippocampus correlates inversely with memory test performance in humans. *Human Brain Mapping*, 34(11):3066–3074, 2013.
- H. Ishwaran and J. S. Rao. Spike and slab variable selection: frequentist and bayesian strategies. 2005.
- D. Lambert. Zero-inflated poisson regression, with an application to defects in manufacturing. *Technometrics*, 34(1):1–14, 1992.
- M. C. Murphy, K. C. Chan, S.-G. Kim, and A. L. Vazquez. Macroscale variation in resting-state neuronal activity and connectivity assessed by simultaneous calcium imaging, hemodynamic imaging and electrophysiology. *Neuroimage*, 169:352–362, 2018.
- N. D. L. N. Fusi, C. Lippert and O. Stegle. Warped linear mixed models for the genetic analysis of transformed phenotypes. *Nature Communications*, 5, 2014. doi: doi.org/10.1038/ncomms5890.
- F. Nathoo, A. Babul, A. Moiseev, N. Virji-Babul, and M. Beg. A variational bayes spatiotemporal model for electromagnetic brain mapping. *Biometrics*, 70(1):132–143, 2014.

- N. Opel, J. Goltermann, M. Hermesdorf, K. Berger, B. T. Baune, and U. Dannlowski. Cross-disorder analysis of brain structural abnormalities in six major psychiatric disorders: A secondary analysis of mega- and meta-analytical findings from the enigma consortium. *Biological Psychiatry*, 88(9):678–686, 2020. ISSN 0006-3223. doi: <https://doi.org/10.1016/j.biopsych.2020.04.027>. URL <https://www.sciencedirect.com/science/article/pii/S0006322320315857>. New Mechanisms of Psychosis: Clinical Implications.
- J. Ren, S. Tapert, C. C. Fan, and W. K. Thompson. A semi-parametric bayesian model for semi-continuous longitudinal data. *Statistics in Medicine*, 41(13):2354–2374, 2022.
- S. Q. K. L. M. S. S. Zhao, W. Crouse and X. He. Adjusting for genetic confounders in transcriptome-wide association studies improves discovery of risk genes of complex traits. *Nature Genetics*, 56, 2024. doi: 10.1038/s41588-023-01648-9.
- E. Uffelmann, Q. Q. Huang, N. S. Munung, J. De Vries, Y. Okada, A. R. Martin, H. C. Martin, T. Lappalainen, and D. Posthuma. Genome-wide association studies. *Nature Reviews Methods Primers*, 1(1):59, 2021.
- S. Whitfield-Gabrieli, C. Wendelken, A. Nieto-Castañón, S. K. Bailey, S. A. Anteraper, Y. J. Lee, X.-Q. Chai, D. R. Hirshfeld-Becker, J. Biederman, L. E. Cutting, and et al. Association of intrinsic brain architecture with changes in attentional and mood symptoms during development. *JAMA Psychiatry*, 77(4):378, 2020. ISSN 2168-622X. doi: 10.1001/jamapsychiatry.2019.4208.
- K. K. Yau, K. Wang, and A. H. Lee. Zero-inflated negative binomial mixed regression modeling of over-dispersed count data with extra zeros. *Biometrical Journal: journal of mathematical methods in biosciences*, 45(4):437–452, 2003.
- C.-H. Yu, R. Prado, H. Ombao, and D. Rowe. A bayesian variable selection approach yields improved detection of brain activation from complex-valued fmri. *Journal of the American Statistical Association*, 113(524):1395–1410, 2018.
- Z. Zeng, M. Li, and M. Vannucci. Bayesian image-on-scalar regression with a spatial global-local spike-and-slab prior. *Bayesian Analysis*, 1(1):1–26, 2022.
- F. Zhang, W. Chen, Z. Zhu, Q. Zhang, M. F. Nabais, T. Qi, I. J. Deary, N. R. Wray, P. M. Visscher, A. F. McRae, and J. Yang. Oisca: a tool for omic-data-based complex trait analysis. *Genome Biology*, 20(1):107, 2019. doi: 10.1186/s13059-019-1718-z. URL <https://doi.org/10.1186/s13059-019-1718-z>.

Reviewers' comments:

Reviewer #1 (Remarks to the Author):

Thank you for revising the manuscript. I only have two minor comments/suggestions. I can only reiterate that I found it was a scientifically sound article, presenting useful and novel method in neuroimaging, and which applies them to the ABCD data.

1) I found the sentence below misleading as GCTA can provide SE of the FVE estimates, which may be used to build confidence intervals and quantify uncertainty around the estimate. It may be a formulation issue, but I would tone it down and explain it better.

"In contrast, GCTA lacks the capability to quantify uncertainty for FVE."

2) The figure legends would benefit from being longer, to make the figures almost self explanatory. For example, define all terms and abbreviations and clarify what the bars represents (CI, SD?). The titles such as "Monte Carlo Simulation Result of point estimation." could be a lot more informative, about what is actually aimed to be showed.

Reviewer #2 (Remarks to the Author):

Summary:

In the response from Ren et. al to my previous concerns regarding the unfavorable performance of the proposed methods in simulation studies, the authors extended the previous simulation studies by including a very ideal situation (i.e., independent & normally distributed features), which is impossible to be real in practice in neuroimaging studies (where the features are definitely correlated), but still failed to show that they could provide accurate estimation and inferences, especially for the PNN. Without further improving the method, the authors decided to conclude in this revision that this is a limitation and that the method needs further development in the future. In my opinion, if a method fails to provide valid estimation and inference about a quantity, the method should not claim that it has the advantage of estimating it – the estimation is wrong anyway. I enumerate my concerns about the model's performance in more detail below.

My previous concerns regarding the model performance are not relieved after this revision. If this statistical method paper is still considered for publication even given these obvious problems about the method's performance, then it is critical to at least revise the writing thoroughly to objectively reflect them to the audience. I saw the authors have made some edits on these in the manuscript, but it still needs further work. I listed some suggestions below.

Major comments:

- The point estimation of PNN by ZIV and ZIVM:

- o First, the settings of the newly added simulation studies are too ideal -- the simulated features were assumed to be independent. Such simulation settings already undermine the credibility of the model's performance in real data as they are far away from reality. Moreover, in this ideal situation, the model still fails to provide accurate point estimates for PNN (second row of Figure 2A, right panel).

- o Second, from the simulation studies based on ABCD data, which is closer to real situations as the correlations between the features were maintained, the estimation of the PNN from the ZIV was just completely off (second row of Figure 2B).

- o So in summary, the simulation studies showed that the proposed method is not capable of providing accurate point estimates of PNN.

- The coverage of the proposed CI for FVE and PNN (Figure 3). Without an accurate point estimation for PNN, it's kind of pointless to move forward to evaluate the performance of the corresponding CI. Nevertheless,

- o under all scenarios (either the ones with independent features or ABCD data), the CIs for FVE and PNN are still conservative (i.e., above the nominal level), although now it's harder to notice in figures

than in the tables used by the authors before. The CI from ZIVM for FVE under the setting of independent features may be an exception.

o However, these are essentially still the same results that concerned me in the previous round of review. The authors claimed in the response letter that "Specifically, we note that the CI coverage for FVE and PNN reaches the nominal level as sample size increases in new simulations utilizing independent Gaussian features.", I don't think this description is accurate.

- The newly added simulation studies didn't relieve my previous concerns about the model performance. Given the current concerns about the model's performance from the simulation studies, if the editor and authors decide to move forward with them, then I suggest objectively reflecting them in the manuscript. I listed some suggestions below, they may not be complete, but I hope they are helpful.

o In the abstract (or any place in the manuscript that summarizes the performance of this method), the authors should not say they can "robustly estimate" it but should directly state the actual performance (especially for PNN estimation).

o The second to the last paragraphs in the introduction: if a statistical method fails to accurately estimate PNN, we may not want to say it can estimate it. Maybe consider adding a sentence about the model performance on estimating PNN here.

o Discussion, "Moreover, the ZIV model offers the advantage of providing PNN estimation, enabling uncertainty quantification with credible intervals, ...": as I mentioned above, there is no point in selling the estimation of PNN as an advantage as the estimation is not reliable anyway.

o Section 3.1.2, "When utilizing ABCD tfMRI features to generate outcomes, the coverage rates of the CI consistently exceed the nominal level for different true FVE and PNN": Maybe explicitly saying what "exceeding the nominal level" means would help the readers, e.g., "which means the CIs are wider than expected and less powerful ...".

o Section 3.2: mention the possible under-/over- estimation of PNN when talking about the PNN estimates from ABCD. Same for Discussion.

Minor comments:

- Section 2.7, "The non-null effects were generated from a normal distribution with mean 0 and standard deviation of 0.1.": non-null effect from a distribution with mean 0?

- Discussion, "Despite these challenges, it is important to highlight that the ZIV model, when compared to existing methodologies such as GCTA, offers superior performance.": I would suggest calling them "limitations" instead of "challenges".

- Section 3.1.1, "Given the analogous performance of ZIV and ZIVM for scenarios where the observation count exceeds the feature count, only the ZIV model was implemented and compared with GCTA in simulations based on ABCD tfMRI features." It's interesting to see that the PNN estimated by ZIVM performed better than the ones estimated from ZIV in the independent features setting (with zero-inflated outcomes). I suggest adding ZIVM to the simulation studies shown in Fig 2B, i.e., the ones simulated from ABD with correlated features.

- Section 3.1.1., "ZIV marginally underestimates FVE by approximately 0.02, while GCTA dramatically underestimates it by 50% value.": put the numbers on the same scale for comparison.

- Discussion: "The FVE underestimation is linked to the zero-inflation censoring, leading to incomplete information, while the PNN inconsistency arises from the high correlation among tfMRI features, necessitating wider credible interval to address this uncertainty.": the wider credible interval and the inconsistency of the point estimation are two different problems. It doesn't make sense to me how wider CIs are needed to address the consistency issue in estimation.

Response to Reviewers' comments on the manuscript "Estimating the Total Variance Explained by Whole-Brain Imaging for Zero-inflated Outcomes"

Junting Ren, Robert Loughnan, Bohan Xu, Wesley K. Thompson and Chun Chieh Fan

Responses to Reviewer #1

Overall opinion

Thank you for revising the manuscript. I only have two minor comments/suggestions. I can only reiterate that I found it was a scientifically sound article, presenting useful and novel method in neuroimaging, and which applies them to the ABCD data.

Response: Thank you for your thoughtful review and for recognizing the scientific soundness and novelty of our article. We are encouraged by your feedback and valuable insight. In this second revision, we have updated the manuscript according to your comments, as detailed below.

Minor Comments

(1) *I found the sentence below misleading as GCTA can provide SE of the FVE estimates, which may be used to build confidence intervals and quantify uncertainty around the estimate. It may be a formulation issue, but I would tone it down and explain it better. "In contrast, GCTA lacks the capability to quantify uncertainty for FVE."*

Response: Thank you for pointing out the potential misinterpretation regarding the capabilities of GCTA. Since we can highlight the uncertainty quantification by ZIV without the need for mentioning the formalization of REML-AI used by GCTA, we removed the corresponding statement for clarity and accuracy.

(2) *The figure legends would benefit from being longer, to make the figures almost self explanatory. For example, define all terms and abbreviations and clarify what the bars represents (CI, SD?). The titles such as "Monte Carlo Simulation Result of point estimation." could be a lot more informative, about what is actually aimed to be showed.*

Response: Thank you for your constructive suggestion. We agree that enhancing the figure legends will make the figures more informative and largely self-explanatory. Accordingly, we have expanded figure legends:

Fig. 1 ZIV schematic. A. The histogram of a zero-inflated outcome. The data is highly concentrated at 0 with a long right tail. B. ZIV assumes a latent outcome and the input imaging features, such as region-of-interest measures, have a linear relationship. C. Through variational Bayes algorithm, ZIV estimates both total signal profiles and the local feature characteristics simultaneously. The resulting posterior weights can be used for predictions and feature selection.

Fig. 2 Monte Carlo Simulation Result of point estimation for FVE, PNN and MAE under different types of outcomes, design matrices, number of sample size and true FVE/PNN values. The dashed lines display the true values. Error bars extend to one standard deviation both above and below the mean of the point estimates. For each simulation setup, a total of 200 instances

are conducted to determine the mean and standard deviation of the point estimates. Panel A is simulated using independent standard normal features with true FVE, PNN and number of features fixed at 0.5, 0.1 and 400, where the distributions of the point estimates are shown as a function of sample size and outcome characteristics. Panel B is simulated using ABCD tfMRI image features with 8893 subjects and 885 features. While the sample size and the number of features are fixed, the distributions of the point estimates are shown as a function of true PNN and FVE. FVE: fraction of variance explained; PNN: proportion of non-null; MAE: mean absolute error; LogTime: natural log of computational time in seconds; ZIVM: ZIV model estimated using MCMC algorithm.

Fig. 3 Monte Carlo Simulation Result of CI coverage rate and range for FVE and PNN under different types of outcomes, design matrices, number of sample size and true FVE/PNN values. The dashed lines display the nominal coverage rate of 95%. Error bars extend to one standard deviation both above and below the mean of the range. For each simulation setup, a total of 200 instances are conducted to determine the mean and standard deviation of the coverage rate and range. Panel A is simulated using independent standard normal features with true FVE, PNN and number of features fixed at 0.5, 0.1 and 400. Panel B is simulated using ABCD tfMRI image features with 8893 subjects and 885 features. CI: credible interval.

Fig. 4 Monte Carlo Simulation Result of feature selection sensitivity and false discovery rate using estimated global PNN and local FNNP under different types of outcomes, design matrices, number of sample size and true FVE/PNN values. The dashed lines display the position of 100% on y-axis. Error bars extend to one standard deviation both above and below the mean. For each simulation setup, a total of 200 instances are conducted to determine the mean and standard deviation of the sensitivity and false discovery rate. Panel A is simulated using independent standard normal features with true FVE, PNN and number of features fixed at 0.5, 0.1 and 400. Panel B is simulated using ABCD tfMRI image features with 8893 subjects and 885 features. FDR: false discovery rate; PNN: proportion of non-null; FNNP: individual feature non-null probability

Responses to Reviewer #2

Major Comments

In the response from Ren et. al to my previous concerns regarding the unfavorable performance of the proposed methods in simulation studies, the authors extended the previous simulation studies by including a very ideal situation (i.e., independent & normally distributed features), which is impossible to be real in practice in neuroimaging studies (where the features are definitely correlated), but still failed to show that they could provide accurate estimation and inferences, especially for the PNN. Without further improving the method, the authors decided to conclude in this revision that this is a limitation and that the method needs further development in the future. In my opinion, if a method fails to provide valid estimation and inference about a quantity, the method should not claim that it has the advantage of estimating it – the estimation is wrong anyway. I enumerate my concerns about the model's performance in more detail below. My previous concerns regarding the model performance are not relieved after this revision. If this statistical method paper is still considered for publication even given these obvious problems about the method's performance, then it is critical to at least revise the writing thoroughly to objectively reflect them to the audience. I saw the authors have made some edits on these in the manuscript, but it still needs further work. I listed some suggestions below.

The point estimation of PNN by ZIV and ZIVM:

- First, the settings of the newly added simulation studies are too ideal – the simulated features were assumed to be independent. Such simulation settings already undermine the credibility of the model's performance in real data as they are far away from reality. Moreover, in this ideal situation, the model still fails to provide accurate point estimates for PNN (second row of Figure 2A, right panel).*
- Second, from the simulation studies based on ABCD data, which is closer to real situations as the correlations between the features were maintained, the estimation of the PNN from the ZIV was just completely off (second row of Figure 2B).*
- So in summary, the simulation studies showed that the proposed method is not capable of providing accurate point estimates of PNN.*

The coverage of the proposed CI for FVE and PNN (Figure 3). Without an accurate point estimation for PNN, it's kind of pointless to move forward to evaluate the performance of the corresponding CI. Nevertheless,

- under all scenarios (either the ones with independent features or ABCD data), the CIs for FVE and PNN are still conservative (i.e., above the nominal level), although now it's harder to notice in figures than in the tables used by the authors before. The CI from ZIVM for FVE under the setting of independent features may be an exception.*
- However, these are essentially still the same results that concerned me in the previous round of review. The authors claimed in the response letter that “Specifically, we note that the CI coverage for FVE and PNN reaches the nominal level as sample size increases in new simulations utilizing independent Gaussian features.”, I don't think this description is accurate.*

Response: We appreciate the reviewer's keen critic on our methods, pinpointing the issues on the PNN estimation. As the simulations highlight a bias in the point estimates of PNN, we agree that our manuscript should communicate these limitations more explicitly. We also realize that we should elaborate more on the broader context and utility of the approach, ensuring that we maintain a balanced perspective on both its strengths and limitations. Therefore, based on the suggestions by the reviewer, we revised our manuscript to reflect upon the challenges and opportunities in modeling PNN for brain imaging data.

-
- **Acknowledgment of the ideal simulation setting and bias and over-coverage of CI shown in PNN estimation:** We recognize the bias in PNN estimations especially within the context of neuroimaging studies, where features are inherently correlated. Consequently, we have made adjustments to the manuscript to temper the claims of our method’s capabilities in these scenarios.
 - In the result section: *“Even though the PNN estimation exhibits bias for zero-inflated outcomes, both the ZIV and ZIVM estimation of PNN displayed a trend of converging to the true value as sample size increases as shown in the second row, whereas the GCTA does not provide estimation for PNN.”*
 - In the discussion section: *“The ZIV model has been validated with comprehensive simulations. When simulating zero-inflated outcomes with independent features, we observe that point estimates FVE and PNN progressively align with the ground truth as the sample size increases. However, we do note that the PNN still exhibits bias and the simulation setup is rather ideal and needs to be interpreted with caution.”*
 - For Figure 2B, we acknowledged the biased PNN estimation: *“ZIV’s PNN estimates show a tendency to overestimate at lower true PNN values and underestimate at higher ones.”*
 - Addressing the claim of CI in the result section: *“When utilizing ABCD tfMRI features to generate outcomes, the coverage rates of the CI consistently exceed the nominal level for different true FVE and PNN, indicating that the CIs are wider than expected, which reduces the power of the CIs. Nevertheless, the CI sustains coverage rates above the nominal level across different setups and provides valid inference within a relatively narrow range.”*
 - **Merits of the Method:** Despite these limitations, our method performs comparably to state-of-the-art models for the traditional linear outcomes in terms of FVE estimation. Notably, when dealing with zero-inflated outcomes, no existing models match our method’s performance. The full coverage rate by our estimated credible intervals of PNN across simulated scenarios is also an indication that our method can provide the bound of the underlying true values, despite the bias in the point estimate.
 - **Utility of PNN Estimations:** While the PNN estimations exhibit bias, they still reliably facilitate the correct selection of variables, as shown in the high sensitivity and low false discovery rate from Figure 4, demonstrating practical utility in various analytical contexts. Our method, though imperfect, provides valuable insights and a foundation for further development.

In response to your suggestions, we have undertaken a revision of the manuscript to ensure that the presentation of our method is balanced, clearly delineating its current strengths and areas for future improvement.

(2) *The newly added simulation studies didn’t relieve my previous concerns about the model performance. Given the current concerns about the model’s performance from the simulation studies, if the editor and authors decide to move forward with them, then I suggest objectively reflecting them in the manuscript. I listed some suggestions below, they may not be complete, but I hope they are helpful.*

- *In the abstract (or any place in the manuscript that summarizes the performance of this method), the authors should not say they can “robustly estimate” it but should directly state the actual performance (especially for PNN estimation).*

-
- *The second to the last paragraphs in the introduction: if a statistical method fails to accurately estimate PNN, we may not want to say it can estimate it. Maybe consider adding a sentence about the model performance on estimating PNN here.*
 - *Discussion, “Moreover, the ZIV model offers the advantage of providing PNN estimation, enabling uncertainty quantification with credible intervals, ...”: as I mentioned above, there is no point in selling the estimation of PNN as an advantage as the estimation is not reliable anyway.*
 - *Section 3.1.2, “When utilizing ABCD tfMRI features to generate outcomes, the coverage rates of the CI consistently exceed the nominal level for different true FVE and PNN”: Maybe explicitly saying what “exceeding the nominal level” means would help the readers, e.g., “which means the CIs are wider than expected and less powerful ...”.*
 - *Section 3.2: mention the possible under-/over- estimation of PNN when talking about the PNN estimates from ABCD. Same for Discussion.*

Response: We sincerely thank the detailed recommendations by the reviewer. Based on the suggestions, we have revised all sections of our manuscript to put the PNN in context:

- **Abstract:** We have revised the abstract to objectively state the model’s capabilities without claiming robust estimation:
“Our zero-inflated variance (ZIV) estimator estimates the fraction of variance explained (FVE) and the proportion of non-null effects (PNN) from large-scale imaging data. In simulations, ZIV demonstrates superior performance over other linear models.”
- **Introduction:** We added this sentence in the introduction to address the concerns about PNN estimation:
“Although the PNN estimates show bias, the combined use of FNNP allows for the selection of true causal features with high sensitivity and a low false discovery rate, as demonstrated in simulations.”
- **Discussion:** We have amended the discussion to reflect the limitations of PNN estimation:
“Moreover, although the estimation of the PNN exhibits bias, the ZIV model effectively utilizes PNN to select important features with high sensitivity and low false discovery rate.”
- **Clarification in Section 3.1.2:** We’ve clarified what exceeding the nominal level means in our manuscript to enhance reader understanding:
“When utilizing ABCD tfMRI features to generate outcomes, the coverage rates of the CI consistently exceed the nominal level for different true FVE and PNN, indicating that the CIs are wider than expected, which reduces the power of the CIs.”
- **Section 3.2 updates:** We have noted the potential bias in PNN estimates:
“However, it is important to note that these PNN estimates may be under or overestimated due to the biases identified in simulations.”

Minor comments

(a) *Section 2.7, “The non-null effects were generated from a normal distribution with mean 0 and standard deviation of 0.1.”: non-null effect from a distribution with mean 0?*

Response: While the non-null effects are generated from a normal distribution with a mean of 0, it’s important to note that the continuous nature of the distribution means that the probability

of an effect being exactly zero is virtually nil. Therefore, the non-null effects, though centered at zero, will not actually be zero.

(b) *Discussion*, “Despite these challenges, it is important to highlight that the ZIV model, when compared to existing methodologies such as GCTA, offers superior performance.”: I would suggest calling them “limitations” instead of “challenges”.

Response: We have revised the terminology and now refer to these as “limitations” instead of “challenges”.

(c) *Section 3.1.1*, “Given the analogous performance of ZIV and ZIVM for scenarios where the observation count exceeds the feature count, only the ZIV model was implemented and compared with GCTA in simulations based on ABCD tfMRI features.” It’s interesting to see that the PNN estimated by ZIVM performed better than the ones estimated from ZIV in the independent features setting (with zero-inflated outcomes). I suggest adding ZIVM to the simulation studies shown in Fig 2B, i.e., the ones simulated from ABD with correlated features.

Response: Previously, we conducted a selective simulations for ZIVM for simulations based on ABCD tfMRI features. In the simulations, we shown that the estimates from ZIV and ZIVM converged even before the sample size reaching to the level of the full ABCD samples. Because the number of outcomes to be tested, ZIVM was too time consuming and did not provide additional accuracies comparing to ZIV. Therefore, we only report the results from the ZIV in the Fig2.

(d) *Section 3.1.1*, “ZIV marginally underestimates FVE by approximately 0.02, while GCTA dramatically underestimates it by 50% value.”: put the numbers on the same scale for comparison.

Response: The sentence has been revised for clarity:

“Across varying true FVE and PNN values, ZIV marginally underestimates FVE by approximately 0.02, while GCTA drastically underestimates it by around 0.10, 0.15 and 0.30 when the true FVE equals 0.25, 0.5, 0.8, respectively.”

(e) *Discussion*: “The FVE underestimation is linked to the zero-inflation censoring, leading to incomplete information, while the PNN inconsistency arises from the high correlation among tfMRI features, necessitating wider credible interval to address this uncertainty.”: the wider credible interval and the inconsistency of the point estimation are two different problems. It doesn’t make sense to me how wider CIs are needed to address the consistency issue in estimation.

Response: We revised the sentence to ensure better clarity, as the following:

“The FVE underestimation is linked to the zero-inflation censoring, leading to incomplete information, while the PNN inconsistency arises from the high correlation among tfMRI features.”

REVIEWERS' COMMENTS:

Reviewer #2 (Remarks to the Author):

I thank the authors for making efforts to address my concerns. I don't have further comments.